# Rethinking Gauss-Newton for learning over-parameterized models

**Michael Arbel, Romain Menegaux,* and Pierre Wolinski**\*†
Univ. Grenoble Alpes, Inria, CNRS, Grenoble INP, LJK,38000 Grenoble, France
`firstname.lastname@inria.fr`

## Abstract

This work studies the global convergence and implicit bias of Gauss Newton's (GN) when optimizing over-parameterized one-hidden layer networks in the mean-field regime. We first establish a global convergence result for GN in the continuous-time limit exhibiting a faster convergence rate compared to GD due to improved conditioning. We then perform an empirical study on a synthetic regression task to investigate the implicit bias of GN's method. While GN is consistently faster than GD in finding a global optimum, the learned model generalizes well on test data when starting from random initial weights with a small variance and using a small step size to slow down convergence. Specifically, our study shows that such a setting results in a hidden learning phenomenon, where the dynamics are able to recover features with good generalization properties despite the model having sub-optimal training and test performances due to an under-optimized linear layer. This study exhibits a trade-off between the convergence speed of GN and the generalization ability of the learned solution.

## 1 Introduction

Gauss-Newton (GN) and related methods, like the Natural Gradient (NG) can offer faster convergence rates compared to gradient descent due to the improved dependence on the conditioning of the problem [9, 21, 30, 31]. For these reasons, these methods have attracted attention in machine learning as an alternative to gradient descent when optimizing ill-conditioned problems arising from the use of vastly over-parameterized networks and large training sets [29, 45]. Unfortunately, GN's high computational cost per iteration, which involves solving an expensive linear system, restricts its applicability in large-scale deep learning optimization. Addressing this challenge has been a primary focus, with extensive efforts dedicated to crafting computationally efficient approximations for GN/NG methods where the aim is to strike a delicate balance between computational cost and optimization speed, thereby enhancing scalability [4, 9, 11, 21, 31, 42].

While substantial effort has been put into making GN scalable, little is known about the global convergence of these methods and their generalization properties when optimizing neural networks, notably compared to gradient descent. Understanding these properties is particularly relevant knowing that state-of-the-art machine learning methods rely on over-parameterized networks for which infinitely many solutions interpolating the data exist albeit exhibit varying generalization properties that depend on *implicit bias* of the optimization procedure [7, 39]. To date, the majority of studies on generalization in over-parameterized networks have primarily focused on analyzing the dynamics of gradient descent and gradient flows – their continuous-time limit –, making important progress in understanding the implicit bias of these optimization procedures. Specifically, [15]

---

*Equal contribution.
†Now affiliated with the LMO, Université Paris-Saclay, Orsay, France.

37th Conference on Neural Information Processing Systems (NeurIPS 2023).

showed that a gradient flow can run under two regimes which yield solutions with qualitatively different generalization properties: the *kernel regime*, where the internal features of a network remain essentially constant during optimization, often yields poor generalization, and the *feature learning regime*, which allows the internal features of a network to adapt to the data, seems behind the impressive generalization properties observed in practice. In addition, the same work precisely defines conditions for these regimes: The *kernel regime* systematically occurs, for instance, when taking the *Neural Tangent limit* (NTK), i.e. scaling the output of a network of width $M$ by a factor $1/\sqrt{M}$, while the *feature learning regime* can arise when instead taking the *mean-field limit*, i.e. scaling the output by a factor of $1/M$. However, these insights rest upon attaining a global solution, which is far from obvious given the non-convexity of the training objective. While global linear convergence of gradient flows is known in the kernel regime [1, 23], obtaining such convergence guarantees in the *feature learning regime* remains an active research area [8, 10, 12, 14]. Importantly, all these works have focused solely on gradient flows and gradient descent, given their prevalent usage when optimizing neural networks.

In contrast, GN/NG methods have received limited attention in the study of both generalization and global convergence for over-parameterized networks. The added complexity of these methods makes their analysis intricate, particularly in the stochastic setting where inaccurate estimates of the pre-conditioning matrix can alter the dynamics in a highly non-trivial manner [32, Section 12.2]. Recent works studied the convergence and generalization of GN in the kernel regime [11, 20, 26, 52], while others focused on linear networks [27]. However, the behavior of these methods for non-linear over-parameterized models remains understudied in the *feature learning regime* which is a regime of interest as it is more likely to result in good generalization.

**Contributions.** The present work aims to deepen our understanding of the properties of Gauss-Newton (GN) methods for over-parameterized networks, building upon recent advancements in neural network optimization [15]. Unlike prior works that primarily addressed scalability issues of GN/NG or studied the impact of stochastic noise on optimization dynamics, we uniquely focus on studying the convergence and implicit bias of an *exact GN method*. Specifically, our research investigates GN in the context of a regression problem using over-parameterized one-hidden layer networks in a deterministic (*full-batch*) setting. By examining an exact, deterministic GN method, rather than an approximate or stochastic version, we eliminate potential regularization effects stemming from approximation and stochasticity that might influence convergence and generalization. This strategy, in turn, enables us to provide both theoretical and empirical insights, shedding light on the properties of GN methods when optimizing neural networks:

1. We provide a global convergence result in Proposition 3 with a linear rate for the continuous-time limit of GN's dynamics in the over-parameterized regime that holds under the mean-field limit. The achieved rate shows better conditioning than the gradient flow with the same objective, emphasizing the method's advantage from a pure optimization perspective. While similar rates can be achieved under strong non-degeneracy conditions as in [50], these conditions are never met over-parameterized networks. Instead, our result relies on a simple condition on the initial parameters of a similar nature as in [13, Thm 3.3]: (i) that the hidden weights are diverse enough so that an interpolating solution can be obtained simply by fitting the linear weights, and (ii) that the linear weights result from a simple fine-tuning procedure to reach near optimality. The result shows that GN can reach the data interpolation regime, often required for studying the implicit bias of optimization (Section 4).

2. The empirical study in Section 5 complements the convergence result in Proposition 3 by further investigating the implicit bias of GN on a student/teacher regression task using synthetic data. We show that, while GN attains an interpolating solution, as predicted by Proposition 3, such a solution generalizes more or less well depending on the regime under which the GN dynamics is running. Specifically, we show that the GN dynamics can exhibit both kernel and feature learning regimes depending on the choice of step size and variance of the weights at initialization. Our experiments also indicate that a *hidden learning* phenomenon occurs in GN dynamics where hidden features that generalize well are obtained even when both *raw* training and test performances are sub-optimal due to an under-optimized linear layer. Quite surprisingly, these features are found when using small step sizes and, sometimes, generalize better than those learned by gradient descent. This is contrary to the common prescription of using larger step sizes in gradient descent for better generalization [3]. Our results suggest a tradeoff between optimization speed and generalization when using GN which is of practical interest.

## 2 Related work

**Convergence of Gauss-Newton's methods.** A large body of work on inverse problems studied the local convergence of Gauss-Newton and its regularized versions such as the Levenberg-Marquardt method [25, 38]. Many such works focus on a continuous-time analysis of the methods as it allows for a simplified study [24, 41]. In this work, we also consider continuous-time dynamics and leave the study of discrete-time algorithms to future work. More recently, increasing attention was given to global convergence analysis. [50] studied the global convergence of GN with an *inexact oracle* and in a stochastic setting establishing linear convergence. The analysis requires that the smallest singular value of the Jacobian is positive near initialization, a situation that can never occur in the over-parameterized regime. [34] studied the convergence of the Levenberg-Marquardt dynamics, a regularized version of Gauss-Newton's method, in a non-convex setting under a *cubic growth* assumption and showed global albeit sub-linear convergence rates. It is unclear, however, if such *cubic growth* assumption holds in our setting of interest. Closest to our work is [11] which aims to accelerate optimization by solving a kernel regression in the NTK scaling limit. There, the authors show global convergence of a regularized version of GN in the *kernel regime*. We aim to investigate the behavior of GD in the *feature learning* regimes.

**The implicit bias of gradient flows.** In the over-parameterized setting, the training objective often possesses several global minima that perfectly interpolate the training data. The generalization error thus heavily depends on the solution selected by the optimization procedure. [39] highlighted the importance of understanding the properties of the selected solutions for a given initialization and several works studied this problem in the case of linear neural networks [6, 33, 40, 49, 51]. For non-linear networks, there is still no general characterization of the implicit bias in the regression setting, although [10] recently made important progress in the case of a one-hidden layer network with ReLU activation and orthogonal inputs showing that gradient flows select a minimum norm interpolating solution when the initial weights are close to zero. Recently, the implicit bias of gradient flows was shown to play a crucial role in other optimization problems such as non-convex bi-level optimization [5, 47], therefore illustrating the ubiquity of such phenomenon in deep learning. The present work empirically studies the implicit bias of Gauss-Newton which is vastly understudied.

## 3 Setup and preliminaries

### 3.1 Regression using over-parameterized one-hidden layer networks

We consider a regression problem where the goal is to approximate an unknown real-valued function $f^\star$ defined over a subset $\mathcal{X}$ of $\mathbb{R}^d$ from a pair of i.i.d. inputs/outputs $(x_n, y_n)_{1 \le n \le N}$ where each $x_n$ is a sample from some unknown distribution $\mathbb{P}$ and $y_n = f^\star(x_n)$. We assume that $f^\star$ belongs to $L_2(\mathbb{P})$, the set of square-integrable functions w.r.t. $\mathbb{P}$ and note that $f^\star$ always belongs to $L_2(\hat{\mathbb{P}})$, where $\hat{\mathbb{P}}$ denotes the empirical distribution defined by the samples $(x_1, \ldots, x_N)$. For simplicity, we assume the data are non-degenerate, meaning that $x_n \neq x_{n'}$ whenever $n \neq n'$. In this case $L_2(\hat{\mathbb{P}})$ is isomorphic to $\mathbb{R}^N$ (i.e., $L_2(\hat{\mathbb{P}}) \cong \mathbb{R}^N$) and we can identify any function $f \in L_2(\hat{\mathbb{P}})$ with the evaluation vector $(f(x_1), \ldots, f(x_N))$. We are interested in approximating $f^\star$ using a one-hidden layer network $f_w$ with parameter vector $w$ belonging to some, possibly infinite, Hilbert space $\mathcal{W}$. The network's parameters are learned by minimizing an objective of the form $\mathcal{L}(f_w)$ where $\mathcal{L}$ is a non-negative, $L$-smooth, and $\mu$-strongly convex function defined over $L_2(\hat{\mathbb{P}})$ and achieving a 0 minimum value at $f^\star$. A typical example that we consider in Section 5 is the mean-squared error over the training set:

$$\min_{w \in \mathcal{W}} \mathcal{L}(f_w), \qquad \mathcal{L}(f_w) = \frac{1}{2N} \sum_{n=1}^{N} (f_w(x_n) - y_n)^2. \tag{1}$$

For the convergence results, we will consider both *finite-width* one-hidden layer networks and their *mean-field infinite-width limit*. In the experiments, we will restrict to finite-width networks although, we will be using a relatively large number of units $M$ compared to the size of the training data, therefore approximating the mean-field limit.

**Finite-width one-hidden layer networks.** Given a non-polynomial point-wise activation function $\gamma$ and some positive integer $M$, these networks take the form:

$$f_w(x) = \frac{1}{M} \sum_{i=1}^{M} v_i \gamma(u_i^\top x), \qquad w = (v, u) \in \mathcal{W} := \mathbb{R}^M \times \mathbb{R}^{M \times d}, \tag{2}$$

where $v \in \mathbb{R}^M$ is the linear weight vector while $u \in \mathbb{R}^{M \times d}$ is the hidden weight matrix. Popular choices for $\gamma$ include ReLU [37] and its smooth approximation $\texttt{SiLU}(x) = x(1 + e^{-\beta x})^{-1}$ which can be made arbitrary close to ReLU by increasing $\beta$ [19]. The above formulation can also account for a bias term provided the input vector $x$ is augmented with a non-zero constant component.

**Mean-field infinite-width limit.** We consider some base probability $\mu$ measure with full support on $\mathcal{X}$ and finite second moment and denote by $L_2(\mu)$ and $L_2(\mu, \mathbb{R}^d)$ the set of square integrable functions w.r.t. $\mu$ and taking values in $\mathbb{R}$ and $\mathbb{R}^d$. Given a non-polynomial point-wise non-linearity, we define infinitely-wide one-hidden layer networks $f_w$ as:

$$f_w(x) = \int v(c) \gamma(u(c)^\top x) \, d\mu(c), \qquad w := (v, u) \in \mathcal{W} := L_2(\mu) \times L_2(\mu, \mathbb{R}^d). \tag{3}$$

Functions of the form (3) correspond to the *mean-field limit* of the network defined in (10) when the number of units $M$ grows to infinity and appear in the Lagrangian formulation of the Wasserstein gradient flow of infinitely wide networks [48]. Existing global convergence results of GN methods were obtained in the context of the NTK limit [11] which does not allow *feature learning*. The mean-field limit we consider here can result in optimization dynamics that exhibit *feature learning* which is why we consider it in our global convergence result analysis. For completeness, we provide a brief discussion on mean-field vs NTK limits and kernel vs feature learning regimes in Appendix A.

**Over-parameterization.** We consider an over-parameterized setting where the network has enough parameters to be able to fit the training data exactly, thus achieving a 0 training error. To this end, we define the gram matrix $G$ which is an $N \times N$ matrix taking the following forms depending on whether we are using a finite-width network ($G^F$) or infinite-width network ($G^I$):

$$G_{n,n'}^F(u) = \frac{1}{M} \sum_{i=1}^{M} \gamma(u_i^\top x_n) \gamma(u_i^\top x_{n'}), \quad G_{n,n'}^I(u) = \int \gamma(u(c)^\top x_n) \gamma(u(c)^\top x_{n'}) \, d\mu(c).$$

We say that a network is over-parameterized if $G(u_0)$ is invertible for some hidden-layer weight $u_0$:

 (A) **(Over-parameterization)** $G(u_0)$ is invertible for an initial parameter $w_0 = (v_0, u_0) \in \mathcal{W}$.

When $u_0$ is sampled randomly from a Gaussian, a common choice to initialize a neural network, Assumption (A) holds for infinitely wide networks as long as the training data are non-degenerate (see [18, Theorem 3.1] for ReLU, and [17, Lemma F.1] for analytic non-polynomial activations). There, the *non-degeneracy* assumption simply means that the inputs are not parallel to each other (i.e. $x_i \nparallel x_j$ whenever $i \neq j$ for any $1 \leq i, j \leq N$), a property that holds almost surely if the data distribution $\mathbb{P}$ has a density w.r.t. Lebesgue measure. In the case of finite-width networks with $M > N$, the result still holds with a high probability when $u$ is sampled from a Gaussian [28].

## 3.2 Generalized Gauss-Newton dynamics

To solve (1), we consider optimization dynamics based on a generalized Gauss-Newton method. To this end, we denote by $J_w$ the Jacobian of $(f_w(x_1), ..., f_w(x_N))$ w.r.t. parameter $w$ which can be viewed as a linear operator from $\mathcal{W}$ to $\mathbb{R}^N$. Moreover, $\nabla \mathcal{L}(f_w)$ denotes the vector of size $N$ representing the gradient of $\mathcal{L}$ w.r.t. the outputs $(f_w(x_1), ..., f_w(x_N))$. We can then introduce the Gauss-Newton vector field $\Phi : \mathcal{W} \to \mathcal{W}$ defined as:

$$\Phi(w) := (J_w^\top H_w J_w + \varepsilon(w) I)^{-1} J_w^\top \nabla \mathcal{L}(f_w),$$

where $H_w$ is a symmetric positive operator on $L_2(\hat{\mathbb{P}})$ and $\varepsilon(w)$ is an optional non-negative (possibly vanishing) damping parameter. Starting from some initial condition $w_0 \in \mathcal{W}$ and for a given positive step-size $\gamma$, the Gauss-Newton updates and their continuous-time limit are given by:

$$w_{k+1} = w_k - \gamma \Phi(w_k), \qquad \dot{w}_t = -\Phi(w_t). \tag{4}$$

The continuous-time limit is obtained by taking the step-size $\gamma$ to $0$ and rescaling time appropriately. When $H_w$ is given by the Hessian of $\mathcal{L}$, the dynamics in (4) recovers the generalized Gauss-Newton dynamics in continuous time when no damping is used [44] and recovers the continuous-time Levenberg-Maquardt dynamics when a positive damping term is added [35]. When the matrix $H_w$ is the identity $H_w = I$, the resulting dynamics is tightly related to the natural gradient [2, 30]. More generally, we only require the following assumption on $H_w$:

(**B**)  $H_w$ is continuously differentiable in $w$ with eigenvalues in $[\mu_H, L_H]$ for positive $L_H, \mu_H$.

**Optional damping.** Amongst possible choices for the damping, we focus on $\varepsilon(w)$ of the form:

$$\varepsilon(w) = \alpha\sigma^2(w) \tag{5}$$

where $\alpha$ is a non-negative number $\alpha \geq 0$, and $\sigma^2(w)$ is the smallest eigenvalue of the *Neural Tangent Kernel* (NTK) $A_w := J_w J_w^\top$ which is invertible whenever $J_w$ is surjective. This choice allows us to study the impact of scalar damping on the convergence rate of the GN dynamics. While computing $\sigma^2(w)$ is challenging in practice, we emphasize that the damping is only optional and that the case when no damping is used, i.e. when $\alpha = 0$, is covered by our analysis.

## 4   Convergence analysis

### 4.1   Global convergence under no-blow up

We start by showing that the continuous-time dynamics in (4) converge to a global optimum provided that it remains defined at all times.

**Proposition 1.** *Let $w_0 = (v_0, u_0) \in \mathcal{W}$ be an initial condition so that $u_0$ satisfies Assumption (**A**). Under Assumption (**B**) and assuming that the activation $\gamma$ is twice-continuously differentiable, there exists a unique solution to the continuous-time dynamics in (4) defined up to some maximal, possibly infinite, positive time $T$. If $T$ is finite, we say that the dynamics **blows-up** in finite time. If, instead $T$ is infinite, then $f_{w_t}$ converges to a global minimizer of $\mathcal{L}$. Moreover, denoting by $\mu$ the strong convexity constant of $\mathcal{L}$ and defining $\mu_{GN} := 2\mu/(L_H(1 + \alpha/\mu_H))$, it holds that:*

$$\mathcal{L}(f_{w_t}) \leq \mathcal{L}(f_{w_0})e^{-\mu_{GN}t}. \tag{6}$$

Proposition 1 is proved in Appendix B.2. It relies on the Cauchy-Lipshitz theorem to show that the dynamics are well-defined up to a positive time $T$ and uses strong convexity of $\mathcal{L}$ to obtain the rate in (6). The result requires a smooth non-linearity, which excludes ReLU. However, it holds for any smooth approximation to ReLU, such as SiLU. As we discuss in Section 5, this does not make a large difference in practice as the approximation becomes tighter. The rate in (6) is only useful when the dynamics is defined at all times which is far from obvious as the vector field $\Phi$ can diverge in a finite time causing the dynamics to explode. Thus, the main challenge is to find conditions ensuring the vector field $\Phi$ remains well-behaved which is the object of Section 4.2.

**Comparaison with gradient flow.** The rate in (6) only depends on the strong-convexity constant $\mu$ of $\mathcal{L}$, the smoothness constant of $H$, and damping strength $\alpha$, with the fastest rate achieved when no damping is used, i.e. $\alpha = 0$. For instance, when choosing $H$ to the identity and $\alpha = 0$, the linear rate becomes $\mu_{GN} = 2\mu$. This is in contrast with a gradient flow of $w \mapsto \mathcal{L}(f_w)$ for which a linear convergence result, in the kernel regime, follows from [15, Theorem 2.4]:

$$\mathcal{L}(f_{w_t}) \leq \mathcal{L}(f_{w_0})e^{\mu_{GD}t},$$

with $\mu_{GF} := \mu\sigma(w_0)/4$ where $\sigma(w_0)$ is the smallest singular value of the initial NTK matrix $A_{w_0}$. In practice, $\mu_{GF} \ll \mu_{GN}$ since the linear rate $\mu_{GF}$ for a gradient flow is proportional to $\sigma(w_0)$, which can get arbitrarily small as the training sample size $N$ increases thus drastically slowing down convergence [43].

### 4.2   Absence of blow-up for almost-optimal initial linear layer

In this section, we provide a condition on the initialization $w_0 = (v_0, u_0) \in \mathcal{W}$ ensuring the Gauss-Newton dynamics never blows up. We start with a simple result, with a proof in Appendix B.1, showing the existence of a vector $v^\star$ so that $w^\star := (v^\star, u_0) \in \mathcal{W}$ interpolates the training data whenever the over-parameterization assumption Assumption (**A**) holds for a given $u_0$.

**Proposition 2.** *Let $w_0 = (v_0, u_0) \in \mathcal{W}$ be an initial condition so that $u_0$ satisfies Assumption (A). Define $\Gamma(u) := \left(\gamma(u^\top x_n)\right)_{1 \leq n \leq N}$ and $F^\star := (f^\star(x_1), ..., f^\star(x_N))$ and let $v^\star$ be given by:*

$$v^\star = \Gamma(u_0)G(u_0)F^\star.$$

*Then the vector $w^\star = (v^\star, u_0) \in \mathcal{W}$ is a global minimizer of $w \mapsto \mathcal{L}(f_w)$ and a fixed point of the Gauss-Newton dynamics, i.e. $\Phi(w^\star) = 0$.*

The above result ensures a global solution can always be found by optimizing the linear weights while keeping the hidden weights fixed. In practice, this can be achieved using any standard optimization algorithm such as gradient descent since the function $v \mapsto l(v) = \mathcal{L}(f_{(v,u_0)})$ converges to a global optimum as it satisfies a Polyak-Lojasiewicz inequality (see Proposition 4 in Appendix B.1). However, this procedure is not of interest here as it does not allow us to learn the hidden features. Nonetheless, since $w^\star = (v^\star, u_0)$ is a fixed point of the Gauss-Newton dynamics, we can expect that an initial condition $w_0 = (v_0, u_0)$, where $v_0$ is close to $v^\star$, results in a well-behaved dynamics. The following proposition, proven in Appendix B.4, makes the above intuition more precise.

**Proposition 3.** *Let $w_0 = (v_0, u_0) \in \mathcal{W}$ be an initial condition so that Assumption (A) holds, i.e. the gram matrix $G(u_0)$ is invertible, and denote by $\sigma_0^2$ the smallest eigenvalue of $G(u_0)$. Assume that the activation $\gamma$ is twice-continuously differentiable and that Assumption (B) holds. Let $R$ be any arbitrary positive number and define $C_R = \sup_{w \in \mathcal{B}(w_0, R)} \|\partial_w J_w\|_{op}$ where $\|.\|_{op}$ denotes the operator norm. If the linear layer is almost optimal $v_0$ in the sense that:*

$$\|\nabla\mathcal{L}(f_{w_0})\| < \epsilon, \qquad \epsilon := (\mu\mu_H\mu_{GN}/8LN)\min(R, C_R^{-1})\min\left(\sigma_0, \sigma_0^2\right), \tag{7}$$

*then (4) is defined at all times, i.e. $T = +\infty$, the objective $\mathcal{L}(f_{w_t})$ converges at a linear rate to 0 according to (6) and the parameters $w_t$ remain within a ball of radius $R$ centered around $w_0$.*

Proposition 3 essentially states that the dynamics never blow up, and hence, converge globally at the linear rate in (6) by Proposition 1, provided the hidden features are diverse enough and the initial linear weights optimize the objective well enough. As discussed in Section 3.1, the first condition on the hidden weights $u_0$ typically holds for a Gaussian initialization when the data are non-degenerate. Additionally, the near-optimality condition on the linear weights $v_0$ can always be guaranteed by optimizing the linear weights while keeping the hidden ones fixed.

**Remark 1.** In the proof of Proposition 3, we show that the occurrence of a finite-time blow-up is tightly related to the Neural Tangent Kernel matrix $A_{w_t}$ becoming singular at $t$ increases which causes the vector field $\Phi$ to diverge. When the NTK matrix $A_{w_t}$ at time $t$ is forced to remain close to the initial one $A_{w_0}$, as is the case in the NTK limit considered in [11], one can expect that the singular values of $A_{w_t}$ remain bounded away from 0 since this is already true for $A_{w_0}$. Therefore, the main technical challenge is the study, performed in Propositions 7 to 9 of Appendix B.3, of the time evolution of the eigenvalues of $A_{w_t}$ in the *mean-field limit*, where $A_{w_t}$ is allowed to differ significantly from the initial NTK matrix $A_{w_0}$. The final step is to deduce that, while the weights $w_t$ are allowed to be in a ball centered in $w_0$ of arbitrarily large radius $R$, the singular values remain bounded away from 0 provided the initial linear weights $v_0$ satisfy the condition in (7).

**Remark 2.** Comparing Proposition 3 to the convergence results for gradient flows in the *kernel regime* [15, Theorem 2.4], our result also holds for an infinite dimensional space of parameters such as in the *mean-field limit* of one-hidden layer network. Moreover, while [15, Theorem 2.4] requires the norm $w_t$ to be within a ball of fixed radius $R_0$ determined by initial smallest singular value $\sigma(w_0)$ of $A_{w_0}$, our result allows $w_t$ to be within arbitrary distance $R$ regardless of the initial $\sigma(w_0)$ provided the condition in (7) holds. Proposition 3 is more similar in flavor to the convergence result of the Wasserstein gradient flow for some probability functionals in [13, Theorem 3.3, Corollary 3.4] which requires the initial objective to be smaller than a given threshold.

While the results of this section guarantee global convergence of the training objective, they do not guarantee learning features that generalize well. Next, we empirically show that GN can exhibit regimes where it yields solutions with good generalization properties.

## 5 Empirical study of generalization for a Gauss-Newton algorithm

We perform a comparative study between gradient descent (GD) and Gauss-Newton (GN) method in the context of over-parameterized networks. Since we observed little variability when changing the

seed, we have chosen to use a single seed for the sake of clarity in the figures when presenting the results of this section, while deferring the analysis for multiple seeds to Appendix C.1. Additional experiments using MNIST dataset [16] are provided in Appendix C.4. The results presented below were obtained by running 720 independent runs, each of which optimizes a network given a specific configuration on a GPU. The total time for all runs was 3600 GPU hours.

## 5.1 Experimental setup

We consider a regression task on a synthetic dataset consisting of $N$ training points $(X_n, Y_n)_{1 \leq n \leq N}$. The objective is to minimize the mean-squared error $\mathcal{L}(f_w)$, defined in (1), over the parameters $w$ of a model $f_w$ that predicts the target values $Y_n$ based on their corresponding input points $X_n$.

**Data generation.** We generate $N$ i.i.d. samples denoted as $(X_n)_{1 \leq n \leq N}$ from a standard Gaussian distribution of dimension $d = 10$. Each corresponding target $Y_n$ is obtained by applying a predefined function $f^\star$, referred to as the *teacher network*, to the input $X_n$ (i.e., $Y_n = f^\star(X_n)$). We choose $f^\star$ to be a one-hidden layer network of the form in (10) with $M^\star = 5$ hidden units $(v_i^\star, u_i^\star)_{1 \leq i \leq M^\star}$ drawn independently from a standard Gaussian distribution. Furthermore, we consider two non-linearities $\gamma$ when defining $f^\star$: the widely used ReLU activation [37] for the main results and its smooth approximation SiLU for additional ablations in Appendix C.3. This choice of target function $f^\star$ results in a hidden low-dimensional structure in the regression task as $f^\star$ depends only on 5-dimensional linear projection of the input data. In most cases, we take the size of the training data to be $N = 500$ except when studying the effect of the training size. This choice allows us to achieve a balance between conducting an extensive hyper-parameter search and being able to compute precise GN updates on the complete training data within a reasonable timeframe. Finally, we generate 10000 test samples to measure the generalization error.

**Model.** We consider a well-specified setting where the model $w \mapsto f_w$, referred to as the *student network*, is also a one-hidden layer with the same activation function $\gamma$ as the *teacher network* $f^\star$. Importantly, the student network possesses a number $M = 5000$ of hidden units, denoted as $w := (v_i, u_i)_{1 \leq i \leq M}$, which is much larger than the teacher's number of units $M^\star = 5$ and allows fitting the training data perfectly (over-parameterized regime). Following [15], we also normalize the output of the student $f_w$ by $M$ to remain consistent with the *mean-field limit* as $M$ increases.

**Initialization.** We initialize the student's hidden units according to a centered Gaussian with standard deviation (std) $\tau_0$ ranging from $10^{-3}$ to $10^3$. Varying the initial std $\tau_0$ allows us to study the optimization dynamics in two regimes: the *kernel regime* (large values of $\tau_0$) and the *feature learning regime* (small values of $\tau_0$). Finally, we initialize the weights of the last layer to be 0. This choice allows us to perform a systematic comparison with the minimum norm least squares solution obtained using random features as described next.

**Methods and baselines.** We consider three optimization methods for the model's parameters: a regularized version of Gauss-Newton (GN), gradient descent (GD), and optimizing the linear layer parameters alone, which can be viewed as a random features (RF) model.
**(GN):** We use the discrete GN updates in (4) with a constant step-size $\lambda$ and $H_w = I$. Each update is obtained using Woodbury's matrix identity by writing $\Phi(w_k) = J_{w_k}^\top z_k$ with $z_k$ the solution of a linear system $\left(J_{w_k} J_{w_k}^\top + \epsilon(w_k) I\right) z_k = \nabla \mathcal{L}(f_{w_k})$ of size $N$. Here, we use the damping defined in (5) with $\alpha = 1$ and ensure it never falls below $\epsilon_0 = 10^{-7}$ to avoid numerical instabilities. In practice, we found that small values of $\epsilon_0$ had little effect on the results (see Figure 4 (Right) of Appendix C.1).
**(GD):** The model's parameters $w$ are learned using gradient descent with a constant step-size $\lambda$.
**(RF):** Instead of optimizing the entire parameter vector $w = (v, u)$ of the model $f_w$, we selectively optimize the parameter $v$ of the linear layer while keeping the weights $u$ of the hidden layer constant. This procedure corresponds to computing a minimal-norm least squares solution $v^{RF}$ for a random features (RF) model, where the features are obtained from the output of the hidden layer at initialization. Specifically, the solution $v^{RF}$ is obtained as

$$v^{RF} = \Gamma(u)(\Gamma(u)^\top \Gamma(u))^\dagger \mathbb{Y}, \tag{8}$$

where $\Gamma(u) = (\gamma(u^\top X_n))_{1 \leq n \leq N} \in \mathbb{R}^{M \times N}$ is the feature vector computed over the training data, and $\mathbb{Y}$ is a vector of size $N$ consisting of the target values $Y_n$. The choice of the un-regularized solution as a baseline is supported by recent findings [22], demonstrating its good generalization in the over-parameterized regime.

**Stopping criterion.** For both (GD) and (GN), we perform as many iterations as needed so that the final training error is at least below $10^{-5}$. Additionally, we stop the algorithm whenever the training error drops below $10^{-7}$ or when a maximum number of iterations of $K^{GD} = 10^6$ iterations for (GD) and $K^{GN} = 10^5$ iterations for (GN) is performed. For (RF) we solve the linear system exactly using a standard linear solver.

## 5.2 Performance metrics

In addition to the training and test losses, we introduce two complementary metrics to assess the quality of the hidden features $\Gamma(u)$ learned by the student model.

**Weighted cosine distance (WCD).** Measuring proximity between the student and teacher's hidden weights $(u_1, ..., u_M)$ and $(u_1^\star, ..., u_{M^\star}^\star)$ requires being able to find a correspondance between each student's parameter $u_i$ and a teacher's one $u_j^\star$. When using a positively homogeneous non-linearity such as ReLU, only the alignment between vectors is relevant. However, since the student's model is vastly over-parameterized, there could be vanishing weights $u_i = 0$ which do not influence on the network's output, while the remaining weights are well aligned with the teacher's weights. We introduce the following weighted distance to account for these specificities:

$$\text{WCD}(u, u^\star) = 2 \sum_{i=1}^M p_i \left( 1 - \max_{1 \le j \le M^\star} \frac{u_i^\top u_j^\star}{\|u_i\| \|u_j^\star\|} \right), \qquad p_i = \frac{\|u_i\|^2}{\sum_{k=1}^M \|u_k\|^2}. \tag{9}$$

Equation (9) finds a teacher's parameter $u_j^\star$ that is most aligned with a given student's parameter $u_i$ and downweights its cosine distance if $u_i$ has a small norm. In practice, we found this measure to be a good indicator for generalization.

**Test loss after linear re-fitting (Test-LRfit).** To evaluate the relevance of the hidden features $\Gamma(u)$, we train a new linear model $\tilde{v}$ on those features and measure its generalization error. In practice, we freeze the weights $u$ of the hidden layer and fit the last layer $v$ using the procedure (8) described above for the random features (RF) baseline. In our over-parameterized setting, this new model should always achieve perfect training loss, but its generalization error strongly depends on how the features were learned.

## 5.3 Numerical results

**Implicit bias of initialization.** Figure 1(Left) shows that GN and GD generalize more or less well compared to a random features model depending on the variance $\tau_0$ of the weights at initialization. First, in the case of the RF model, changing $\tau_0$ barely affects the final training and test error. This is essentially due to the choice of the non-linearity ReLU which is positively homogeneous. Second, for large $\tau_0$, the final test error of both GN and GD matches that of RF suggesting that the dynamics were running under the *kernel regime/lazy regime* [15, 23] where the final layer's weights are learned without changing the hidden features much. Finally, for small values of $\tau_0$, both GN and GD obtained a better test error than RF which can only be explained by learning suitable hidden features. These results are further confirmed when varying the size of the training set as shown in Appendix C.2. In Appendix C.3, we performed the same experiment using SiLU non-linearity instead of ReLU and observed the same transition between the kernel and feature learning regimes. While prior works such as [11] analyzed Gauss-Newton in the kernel learning regime, our results indicate that Gauss-Newton can also exhibit a feature learning regime for small values of $\tau_0$.

**Implicit bias of the step size.** Figure 1(Right) shows that increasing the step size in Gauss-Newton results in features that do not generalize well. This is unlike gradient descent where larger step sizes yield better-performing features[3]. This behavior is first observed in the top figures showing an increasing *weighted cosine distance* (WCD) between the student's hidden weights and the teacher's weights as the step size increases in the case of Gauss-Newton. On the contrary, the distance decreases with larger step sizes in the case of gradient descent, indicating that the algorithm learns better features. This effect is also confirmed when computing the test loss after refitting the linear layer to the training data exactly while keeping the learned hidden weights fixed (see linear re-fitting in Section 5.2). Several works, such as [3, 36], highlight the importance of taking large step-sizes

---

[3]The step size can be taken as large as possible provided the algorithm does not diverge. In our experiments, we found that going beyond a step size of $\lambda = 10^3$ for GD results in a divergence of the algorithm

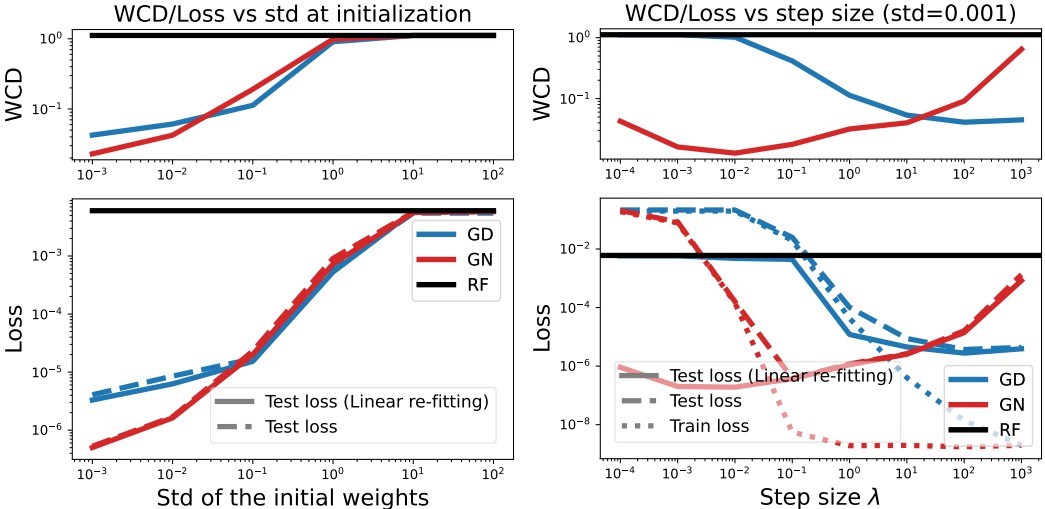

Figure 1: Final values of various metrics vs std of the hidden layer's weights at initialization $\tau_0$ (left) and step size (right) for a ReLU network. (Left figure) The training size is set to $N = 500$ while $\tau_0$ ranges from $10^{-3}$ to $10^2$. For both GD and GN, results are reported for the best-performing step-size $\lambda$ selected according to the test loss on a regular logarithmic grid ranging from $10^{-3}$ to $10^3$. (Right figure) The std of the weights at initialization is set to $\tau_0 = 10^{-3}$. All models are optimized up to a training error of $10^{-6}$ or until the maximum number of steps is exceeded, ($M = 5000$ , $N = 500$).

for gradient descent, to improve generalization, our results show that Gauss-Newton benefits instead of using smaller step-sizes. As shown in Appendix C.3, this behavior persists when using a different non-linearity, such as SiLU with different values for $\beta$ (1 and $10^6$). Interestingly, we observed that GN might underperform GD as the parameter $\beta$ defining SiLU decreases making SiLU nearly linear and less like ReLU. In all cases, while the final test loss remains large for small step sizes (in both GD and GN), the test loss after linear re-fitting is much lower in the case of Gauss-Newton, indicating that the algorithm was able to learn good features while the last layer remained under-optimized.

**Hidden feature learning.** Figure 2 (Left) shows the evolution of various metrics, including the WCD, the training, and test errors with the number of iterations for both GN and GD. For each method, results for the best step size and variance at initialization are presented (($\tau_0, \lambda$) = ($10^{-3}, 10^2$) for GD and ($\tau_0, \lambda$) = ($10^{-3}, 10^{-2}$) for GN). The bottom figure clearly indicates that the early iterations of GN essentially optimize internal features first while barely improving the last layer. This can be seen deduced from the training and test losses which remain almost constant at the beginning of optimization, while the test loss after linear re-fitting steadily decreases with the iterations. The last layer is learned only towards the end of training as indicated by a faster decrease in both train and test objectives. Gradient descent displays a different behavior with test loss before and after linear re-fitting remaining close to each other during optimization.

**Better generalization of GN comes at the cost of a slower training.** Figure 2 (Right) shows the evolution of the same quantities as in the left figures for $\tau_0 = 10^{-3}$ as a function of time (in seconds) and for the three values of the step-size $\lambda$ ($10^{-2}$ (bright colors) 1 (medium colors) and $10^2$ (dark colors)). The figure shows that, while the training loss converges much faster for larger step sizes in the case of GN, it does not result in the best generalization error as measured by the test loss after linear re-fitting. Hence, better generalization comes at the cost of a slower convergence of the training loss. Again, the behavior is quite different in the case of gradient descent for which larger step sizes result in both faster optimization of the training objective and better generalization.

## 6 Conclusion

The results illustrate the importance of the initialization and choice of the step size for the generalization performance of GN when optimizing over-parameterized one-hidden layer networks. While

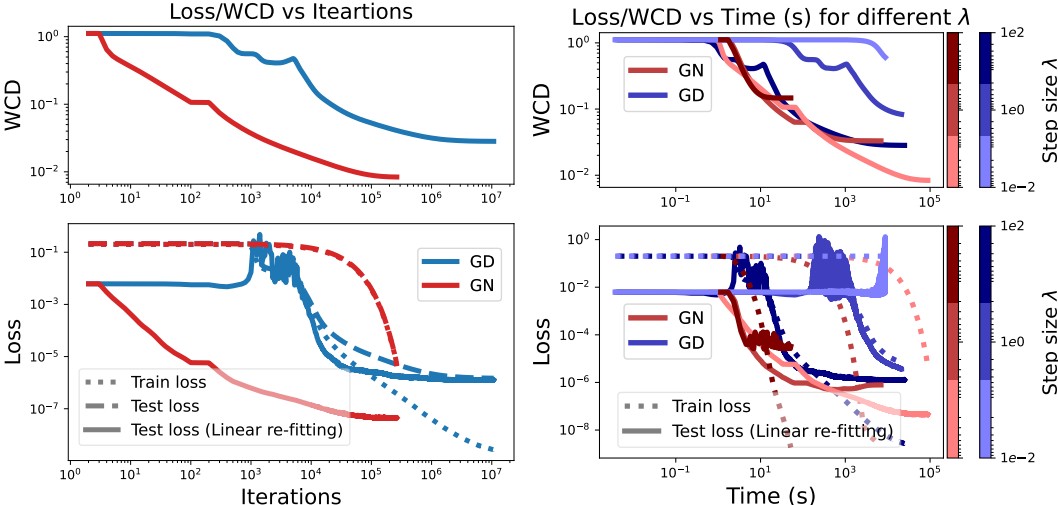

Figure 2: Evolution of various metrics during training for both GD and GN. Both the variance of the weights at initialization $\tau_0$ and step-size $\lambda$ are selected from two sets $\{10^{-3}, 1, 10^2\}$ and $\{10^{-2}, 1, 10^2\}$. In all cases, we report results for the best-performing choice of $\tau_0$ (achieved for $\tau_0 = 10^{-3}$) based on test error after linear re-fitting (Test-LRfit). Both Top-Left and Bottom-Left figures show the evolution of the WCD defined in (9) and the three metrics (the training loss, the test loss and Test-LRfit) for both GD and GN using the step-size $\lambda$ achieving the lowest Test-LRfit. Top-Right and Bottom-Right figures show the evolution of WCD, train loss and test loss after linear re-fitting for the various choices of step-size (darker colors correspond to larger step-sizes).

our theoretical analysis shows that GN can reach a global solution of the training objective when starting from a close-to-optimal initialization, this result does not imply anything about the generalization properties of the obtained solution or the quality of the learned features. Our empirical study instead shows that GN may favor feature learning (thus yielding improved generalization) at the cost of a slower optimization of the training objective due to the use of a smaller step size. Providing a theoretical analysis of these phenomena is an interesting direction for future work. Finally, our study shows that the training or test error may not always be a good indicator of the quality of the learned features due to an under-optimized final layer. Instead, a test/validation error *after re-fitting* the linear layer may be a better indicator for the quality of the learned features which can be of practical use.

## Acknowledgments

The project leading to this work has received funding from the French National Research Agency (ANR-21-JSTM-0001) in the framework of the "Investissements d'avenir" program (ANR-15-IDEX-02). This work was granted access to the HPC resources of IDRIS under the allocation 2023-AD011013762R1 made by GENCI.

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

# A  Background: NTK vs mean-field limits, kernel vs feature learning regimes

We provide a brief discussion of the notions of NTK and mean-field limits as well as the related convergence regimes for a gradient flow. Given a non-polynomial point-wise and differentiable activation function $\gamma$ and some positive integer $M$, consider the following one-hidden network:

$$f_w(x) = \alpha(M) \sum_{i=1}^{M} v_i \gamma(u_i^\top x), \qquad w = (v, u) \in \mathcal{W} := \mathbb{R}^M \times \mathbb{R}^{M \times d}, \tag{10}$$

where $\alpha(M)$ is a scaling factor that depends on the number of neurons $M$. For simplicity, we assume that the weights $(v_i, u_i)$ are i.i.d. samples from a Gaussian. We are interested in the dynamics of a gradient flow when $M$ increases to $+\infty$ and for specific choices of the scaling factor $\alpha(M)$. First, let's introduce some notations. We denote by $F(w)$ the vector of evaluation of the network on the training data $F(w) = (f_w(x_1), \ldots, f_w(x_N))$ and by $J_w$ the Jacobian of $F(w)$ w.r.t. parameter $w$. Moreover, we introduce the NTK matrix $A_w := J_w J_w^\top$ which is an $N \times N$ symmetric positive semi-definite matrix. Finally, $\nabla \mathcal{L}(f_w)$ denotes the vector of size $N$ representing the gradient of $\mathcal{L}$ w.r.t. $F(w)$. The gradient flow of an objective $\mathcal{L}(F_w)$ is given by:

$$\dot{w}_t = -J_{w_t} \nabla_F \mathcal{L}(F_{w_t}),$$

Additionally, it is often informative to track the time evolution of the vector $F_{w_t}$, which is given by the following ODE:

$$\partial_t F_{w_t} = -J_{w_t} J_{w_t}^\top \nabla_F \mathcal{L}(F_{w_t}) = -A_{w_t} \nabla_F \mathcal{L}(F_{w_t}). \tag{11}$$

One can see from (11) that the NTK matrix $A_{w_t}$ arises naturally when expressing the dynamics of the function $F_{w_t}$. We can even provide an explicit expression of $A_{w_t}$ using the NTK kernel, which is a positive semi-definite kernel $K_w(x, x')$ defined by the inner product:

$$K_w(x, x') = \partial_w f(x) \partial_w f(x')^\top = \alpha(M)^2 \sum_{i=1}^{M} \left( \gamma(u_i^\top x) \gamma(u_i^\top x') + (v_i)^2 \gamma'(u_i^\top x) \gamma'(u_i^\top x') x^\top x' \right).$$

The NTK matrix $A_w$ is then formed by collecting the values $(K_w(x_i, x_j))_{1 \le i,j \le N}$ on the training data. It is easy to see that when $\alpha(M) = \frac{1}{\sqrt{M}}$, then $K_w(x, x')$ converges to an expectation under the distribution of the weights $(v_i, u_i)_{1 \le i \le M}$ under mild conditions. In this case, $A_w$ depends on the choice of the distribution of the weights $(v_i, u_i)_{1 \le i \le M}$ and not on the particular samples. On the other hand, when $\alpha(M) = \frac{1}{M}$, it is the function $f_w$ itself that converges to an expectation under the distribution of the weights. This distinction will lead to different qualitative dynamics for the gradient flows as we discuss next.

## A.1  The Neural Tangent Kernel limit and the kernel regime ($\alpha(M) = \frac{1}{\sqrt{M}}$)

When choosing $\alpha(M) = \frac{1}{\sqrt{M}}$ and setting $M \to +\infty$, it is shown in [23] that the NTK matrix $A_{w_t}$ remains constant in time and depends only on the distribution of the weights $w_0$ at initialization. Hence, $F_{w_t}$ is effectively the solution of the ODE:

$$\partial_t F_t = -A_{w_0} \nabla_F \mathcal{L}(F_t), \qquad F_0 = F_{w_0},$$

which converges at a linear rate to a global minimizer of $\mathcal{L}$ whenever $A_{w_0}$ is invertible and $\mathcal{L}$ is strongly convex. The NTK limit results in a kernel regime, where the resulting solution is equivalent to the one obtained using a kernel method with the initial NTK kernel $K_{w_0}(x, x')$ [15, 23].

## A.2  The mean-field limit and the feature learning regime ($\alpha(M) = \frac{1}{M}$)

When choosing instead $\alpha(M) = \frac{1}{M}$, it is possible to view the limiting function as an expectation over weights drawn from some probability distribution $\rho$

$$f_\rho(w) = \int v \gamma(u^\top x) \, d\rho(v, u).$$

In [48], it is further shown that such a formulation is essentially equivalent to the one considered in (3). As a result, the time evolution in (11) does not simplify as in the NTK limit, since the NTK matrix $A_{w_t}$ (and NTK kernel $K_{w_t}$) is allowed to evolve in time. This is a limit in which *feature learning* is possible since the dynamics do not necessarily simplify to a kernel method.

# B Proofs

**Notations.** For an operator $A$ between two Hilbert spaces, denote by $\|A\|_{op}$ its operator norm and by $\|A\|_F$ its Frobenius norm and let $\sigma^\star(A)$ be its smallest singular value whenever it is well-defined. Denote by $\mathcal{B}(w, R)$ the open ball of radius $R$ centered at an element $w$ in a Hilbert space.

## B.1 Over-parameterization implies interpolation.

*Proof of Proposition 2.* Denote by $F^\star = (f^\star(x_1), ..., f^\star(x_n))$. First, it is easy to see that the matrix $\partial_v f_w \partial_v f_w^\top$ is exactly equal to the $N \times N$ matrix $G(u_0)$ considered in Assumption (A), which is assumed to be invertible. Moreover, $\partial_v f_w$ is independent of $v$ since $f_{(v,u_0)}$ is linear in $v$. Hence, we can simply choose:

$$v^\star = (\partial_v f_{(v,u_0)})^\dagger F^\star = (\partial_v f_{(v,u_0)})^\top (\partial_v f_{(v,u_0)} \partial_v f_{(v,u_0)}^\top)^\dagger F^\star = (\partial_v f_{(v,u_0)})^\top G(u_0)^{-1} F^\star.$$

Moreover, using again the linearity of $f$ in $v$, we have that

$$f_{(v^\star,u_0)} = \partial_v f_{(v^\star,u_0)} v^\star = G(u_0) G(u_0)^{-1} F^\star = F^\star.$$

We have therefore shown that $w^\star = (v^\star, u_0)$ is a global minimizer of $w \mapsto \mathcal{L}(f_w)$. Finally, since $\nabla \mathcal{L}(f_{w^\star}) = \nabla \mathcal{L}(f^\star) = 0$, it follows that $\phi(w^\star) = 0$. $\qquad\square$

**Proposition 4.** *Assume that Assumption (A) holds for some vector $u_0$. Then, the function $v \mapsto l(v) = \mathcal{L}(f_{(v,u_0)})$ is convex and satisfies a Polyak-Lojasiewicz inequality:*

$$\|\nabla_u l(u)\|^2 \geq 2\mu\sigma_0^2(l(u) - \mathcal{L}(f^\star)).$$

*with $\mu$ the strong convexity constant of $\mathcal{L}$ and $\sigma_0^2$ the smallest singular value of the matrix $G(u_0)$ appearing in Assumption (A).*

*Proof.* It is easy to see that $l(v)$ is convex as a composition of a strongly convex function $\mathcal{L}$ and a linear function $v \mapsto f_{(v,u_0)}$. We will show now that $l$ satisfies a Polyak-Lojasiewicz (PL) inequality. To this end, we make the following calculations:

$$\|\nabla_u l(u)\|^2 = \nabla_f \mathcal{L}(f_w)^\top \partial_v f_w \partial_v f_w^\top \nabla_f \mathcal{L}(f_w). \tag{12}$$

It is easy to see that the matrix $\partial_v f_w \partial_v f_w^\top$ is exactly equal to the $N \times N$ matrix $G(u_0)$ considered in Assumption (A), which is assumed to be invertible. Hence, its smallest singular value $\sigma_0^2$ is positive and the following inequality holds:

$$\|\nabla_u l(u)\|^2 \geq \sigma_0^2 \|\nabla_f \mathcal{L}(f_w)\|^2. \tag{13}$$

On the other hand, since $\mathcal{L}$ is strongly convex and differentiable, then it satisfies the following PL inequality:

$$\|\nabla_f \mathcal{L}(f_w)\|^2 \geq 2\mu(\mathcal{L}(f_w) - \mathcal{L}(f^\star)), \tag{14}$$

where $\mu$ is the strong convexity constant of $\mathcal{L}$. Hence, we get the desired inequality by combining (13) and (14):

$$\|\nabla_u l(u)\|^2 \geq 2\mu\sigma_0^2(l(u) - \mathcal{L}(f^\star)).$$

$\qquad\square$

## B.2 Global convergence under no blow-up: Proof of Proposition 1

In this section we prove the global convergence result under no blow-up stated in Proposition 1. We start by stating the smoothness assumption on the activation function $\gamma$ as we will be using it in many places.

(C) The non-linearity $\gamma$ is twice-continuously differentiable.

As a first step, we need to prove a local existence and uniqueness of the solution to the ODE in (4) which results from Cauchy-Lipschitz theorem applied to the vector field $\Phi$. To this end, a key step is to show that $\Phi$ is locally Lipschitz near initialization when the Jacobian $J_{w_0}$ is surjective.

**Proposition 5** (Regularity of $\Phi$.)**.** *Assume* **(B)** *and* **(C)** *and that $w_0$ satisfies Assumption* **(A)**. *Then, there exists a neighborhood of $w_0$ so that $\Phi(w)$ is locally Lipschitz in $w$.*

*Proof of Proposition 5.* By application of the Woodbury matrix identity, $\Phi$ can be equivalently written as:

$$\Phi(w) = J_w^\top \left( J_w J_w^\top + \varepsilon(w) H_w^{-1} \right)^{-1} H_w^{-1} \nabla\mathcal{L}(f_w). \tag{15}$$

Moreover, by the smoothness assumption, $J_w$, $H_w$, $\epsilon(w)$ and $\nabla\mathcal{L}(f_w)$ are all differentiable in $w$. Additionally, since $J_{w_0}$ is surjective, it must hold that $\sigma^2(w) > 0$ in a neighborhood of $w_0$ so that $J_w J_w^\top$ remains invertible. Finally, since $H_w$ is always positive, it holds that $\left( J_w J_w^\top + \varepsilon(w) H_w^{-1} \right)^{-1}$ and $H_w^{-1}$ are differentiable in a neighborhood of $w_0$. Therefore, $\Phi$ is Locally Lipschitz in a neighborhood of $w_0$. $\qquad\square$

The next proposition establishes the existence and uniqueness of the dynamics defined in (4).

**Proposition 6** (Local existence)**.** *Under the same conditions as Proposition 5, there exists a unique solution $(w_t)_{t\geq 0}$ to (4) defined up to a, possibly infinite, maximal time $T > 0$.*

Proposition 6 is a direct consequence of the Cauchy-Lipschitz theorem which holds since the vector field $\Phi(w)$ is locally Lipschitz in $w$ in a neighborhood of $w_0$ by Proposition 5. The above local existence result does not exclude situations where the dynamics blow up in finite time $T < +\infty$ and possibly fail to globally minimize the objective $\mathcal{L}$. We will show, later, that finite-time blow-ups never occur under additional assumptions on the initial condition.

*Proof of Proposition 1.* Existence and uniqueness of the continuous-time dynamics up to a maximal time $T > 0$ follows from Proposition 6. It remains to obtain the global convergence rate when $T$ is infinite. For simplicity, we will ignore the dependence on $w_t$ in what follows and will write $b := \nabla\mathcal{L}(f_{w_t})$. By differentiating the objective in time, we get the following expression:

$$\begin{aligned}
\partial_t[\mathcal{L}(f_{w_t})] &= -b^\top J \left( J^\top H J + \varepsilon I \right)^{-1} J b \\
&= -b^\top H^{-1} b + \varepsilon b^\top H^{-1} \left( J J^\top + \varepsilon H^{-1} \right)^{-1} H^{-1} b \\
&= -b^\top H^{-1} b + \varepsilon b^\top H^{-\frac{1}{2}} \left( H^{-\frac{1}{2}} \left( J J^\top + \varepsilon H^{-1} \right)^{-1} H^{-\frac{1}{2}} \right) H^{-\frac{1}{2}} b \\
&\leq -b^\top H^{-1} b \left( 1 - \varepsilon \left\| H^{-\frac{1}{2}} \left( J J^\top + \varepsilon H^{-1} \right)^{-1} H^{-\frac{1}{2}} \right\|_{op} \right) \\
&\leq -b^\top H^{-1} b \underbrace{\left( 1 - \frac{\varepsilon}{\varepsilon + \sigma^\star(H)\sigma^\star(J J^\top)} \right)}_{\geq \eta := (1 + \alpha/\mu_H)^{-1}} \\
&\leq -\eta b^\top H^{-1} b \leq -\eta L_H^{-1} \|b\|^2 \leq -2\eta \mu L_H^{-1} \mathcal{L}(f_{w_t}).
\end{aligned}$$

We used the Woodbury matrix inversion lemma to get the second line while the third and fourth lines use simple linear algebra. To get the fifth line, we used Lemma 3 to upper-bound the operator norm appearing in the fourth line. The last line follows by definition of $\varepsilon(w)$, the fact that $H \leq L_H I$ and the fact that $\frac{1}{2}\|\nabla\mathcal{L}(f)\|^2 \geq \mu\mathcal{L}(f)$ by $\mu$-strong convexity of $\mathcal{L}$. This allows to directly conclude that $\mathcal{L}(f_{w_t}) \leq \mathcal{L}(f_{w_0}) e^{-\mu_{GN} t}$, with $\mu_{GN} := 2\eta\mu L_H^{-1}$. $\qquad\square$

## B.3 Control of the singular values of NTK matrix

In this section, we are interested in the evolution of the smallest singular value of the NTK matrix $A_{w_t} := J_{w_t} J_{w_t}^\top$. For any arbitrary positive radius $R > 0$, define the constant $C_R$ and stopping time $T_R$ as follows:

$$C_R := \sup_{w \in \mathcal{B}(w_0, R)} \|\partial_w J_w\|_{op} < +\infty$$

$$T_R := \sup\{t \geq 0 \quad |\|w_t - w_0\| < R, \text{ and } \sigma^\star(A_{w_t}) > 0\},$$

The stopping time $T_R$ characterizes the smallest-time when the dynamics becomes either degenerate ($\sigma^\star(A_{w_t}) = 0$) or grows too far away from initialization ($w_t \neq \mathcal{B}(w_0, R)$).

To control the singular values of $A_{w_t}$, we find it useful to consider the dynamics of the Frobenius norm of the pseudo-inverse $J_w^\dagger$ which we denote by $a_t := \left\| J_t^\dagger \right\|_F$.

Indeed, simple linear algebra provides the following lower-bound on $\sigma^\star(A_{w_t})$ in terms of $a_t$:

$$\sigma^\star(A_{w_t}) \geq a_t^{-2}. \tag{16}$$

Note that, while the space of parameters can even be infinite-dimensional (mean-field limit), $J_w$ has a finite-dimensional range and thus always admits a pseudo-inverse $J_w^\dagger$ given by:

$$J_w^\dagger = J_w^\top (J_w J_w^\top)^\dagger.$$

The next proposition provides a differential inequality for $a_t = \left\| J_{w_t}^\dagger \right\|_F$ in terms of $\|\dot{w}_t\|$ that holds for all times smaller than $T_R$.

**Proposition 7.** *Assume (B) and (C) and that $w_0$ satisfies Assumption (A), then for any $R > 0$, the time $T_R$ is positive and smaller than the maximal time $T$, i.e. $0 < T_R \leq T$. Moreover, for any $t \in [0, T_R)$, $a_t$ satisfies:*

$$|\dot{a}_t| \leq a_t^2 C_R \|\dot{w}_t\|,$$

*Proof.* **Positive stopping time $T_R$.** By construction, $T_R$ must be smaller than $T$ as it requires $w_t$ to be well-defined. Moreover, $J_{w_0}$ is surjective by assumption and $J_w$ is continuous in $w$, therefore $J_{w_t}$ must also be surjective for small enough positive times $t > 0$. Additionally, it must hold that $\|w_t - w_0\| \leq R$ for small positive times, by continuity of $t \mapsto w_t$. This allows shows that $T_R > 0$.

**Dynamics of $J_{w_t}^\dagger$.** By smoothness of $J_w$ it holds that $J_{w_t}^\dagger$ is differentiable on the open interval $(0, T_R)$ and satisfies the following ODE obtained by direct differentiation:

$$\dot{J}_t^\dagger = -J_t^\dagger \dot{J}_t J_t^\dagger + (I - P_t)\dot{J}_t^\top \left(J_t J_t^\top\right)^{-1}. \tag{17}$$

In (17), we introduced notation $J_t := J_{w_t}$ and its time derivative $\dot{J}_t$ for simplicity and denote by $P_t$, the projector: $P_t := J_t^\top \left(J_t J_t^\top\right)^{-1} J_t$.

**Controlling $|\dot{a}_t|$.** Taking the time derivative of $a_t$ and recalling the evolution (17) yields:

$$a_t \dot{a}_t = -\left(J_t^\dagger\right)^\top J_t^\dagger \dot{J}_t J_t^\dagger.$$

where we used that $\left(J_t^\dagger\right)^\top (I - P_t) = 0$ by Lemma 1 in Appendix B.5. Furthermore, taking the absolute value of the above equation, and recalling that the Frobenius norm is sub-multiplicative for the product of operators (Lemma 4 in Appendix B.5), we directly get:

$$a_t |\dot{a}_t| \leq \left| \left(J_t^\dagger\right)^\top J_t^\dagger \dot{J}_t J_t^\dagger \right| \leq a_t^3 \left\| \dot{J}_t \right\|_F. \tag{18}$$

By the chain rule, we have that $\dot{J}_t = \partial_w J_t \dot{w}_t$ therefore, $\left\| \dot{J}_t \right\|_F \leq \|\partial_w J_t\|_{op} \|\dot{w}_t\|$. Moreover, since $t < T_R$, then it must holds that $w_t \in B(w_0, R)$ so that $\|\partial_w J_t\|_{op} \leq C_R$. Henceforth, $\left\| \dot{J}_t \right\|_F \leq C_T \|\dot{w}_t\|$ and (18) yields the desired inequality after dividing by $a_t$ which remains positive all times $t \in [0, T_R)$ since $J_t$ does not vanish. $\qquad\square$

The next proposition provides an estimate of the time derivatives $|\dot{a}_t|$ and $\|\dot{w}_t\|$ up to time $T_R$. For simplicity, we introduce the notation $\Delta_t = \|\nabla \mathcal{L}(f_{w_t})\|$.

**Proposition 8.** *Assume (B) and (C) and that $w_0$ satisfies Assumption (A). Then, for any $R > 0$, the time $T_R$ is positive and smaller than the maximal time $T$ up to which (4) is defined, i.e. $0 < T_R \leq T$. Moreover, define $C := L\mu^{-1}\mu_H^{-1}\Delta_0 C_R$. Then $a_t$ and $\|\dot{w}_t\|$ satisfy the following differential inequality on $[0, T_R)$:*

$$|\dot{a}_t| \leq C a_t^3 e^{-\frac{\mu_{GN}}{2}t}$$

$$\|\dot{w}_t\| \leq C C_R^{-1} a_t e^{-\frac{\mu_{GN}}{2}t}$$

*Proof of Proposition 8.* By application of Proposition 7 to $(w_t)_{0 \leq t < T}$, we directly have that $T_R$ is positive and smaller than $T$ and the following estimate holds for any $t < T_R$:

$$|\dot{a}_t| \leq a_t^2 C_R \|\dot{w}_t\|,$$

It remains to upper-bound $\|\dot{w}_t\|$.

**Controlling $\|\dot{w}_t\|$.** Ignoring the dependence on $w_t$ in the notations for simplicity and setting $B := H^{\frac{1}{2}} J$, we have, by definition of the dynamical system (4):

$$\begin{aligned}
\|\dot{w}_t\| &\leq \left\| \left( J^\top H J + \varepsilon I \right)^{-1} J^\top \nabla \mathcal{L}(f) \right\| \\
&= \left\| J^\top \left( J J^\top + \varepsilon H^{-1} \right)^{-1} H^{-1} \nabla \mathcal{L}(f) \right\| \\
&= \left\| B^\top \left( B B^\top + \varepsilon \right)^{-1} H^{-\frac{1}{2}} \nabla \mathcal{L}(f) \right\| \\
&\leq \left\| B B^\top \left( B B^\top + \varepsilon I \right)^{-2} \right\|_F^{\frac{1}{2}} \left\| H^{-\frac{2}{2}} \nabla \mathcal{L}(f) \right\| \\
&\leq \left\| B^\dagger \right\|_F \|\nabla \mathcal{L}(f)\| \\
&\leq \mu_H^{-1} \left\| J^\dagger \right\|_F \|\nabla \mathcal{L}(f)\| = \mu_H^{-1} a_t \Delta_t,
\end{aligned}$$

where the second line follows by the Woodbury matrix identity, and the third line follows by simple linear algebra. For the fourth line, we use the properties of the Frobenius norm. The fifth and last lines are direct consequences of Lemmas 2 and 4 in Appendix B.5.

**Concluding.** We can combine the upper-bound on $|\dot{a}_t|$ and $\|\dot{w}_t\|$ to get:

$$|\dot{a}_t| \leq \mu_H^{-1} C_R a_t^3 \Delta_t.$$

Finally, since $t \leq T$, it follows from Proposition 1 that $\Delta_t \leq \frac{L}{\mu} \Delta_0 e^{-\frac{\mu_{GN}}{2} t}$ which allows to conclude. $\square$

The next proposition controls $a_t$ and $\|w_t - w_0\|$ in terms of $T_R$, $C_R$, and other constants.

**Proposition 9.** *Consider the same conditions as in Proposition 8. Let $\zeta := \frac{4L}{\mu \mu_H \mu_{GN}}$ and consider the (possibly infinite) time $T^+$ defined as:*

$$T^+ = -2\mu_{GN}^{-1} \log \left( 1 - (\zeta C_R \Delta_0 a_0^2)^{-1} \right),$$

*with the convention that $T^+ = +\infty$ if $1 \geq \zeta C_R \Delta_0 a_0^2$. Then for any time $t < T^- := \min(T_R, T^+)$ it holds that:*

$$\|w_t - w_0\| \leq \zeta \Delta_0 a_0, \qquad a_t \leq a_0 \left( 1 + \zeta C_R \Delta_0 a_0^2 \left( e^{-\frac{\mu_{GN}}{2} t} - 1 \right) \right)^{-\frac{1}{2}} \tag{19}$$

*Proof of Proposition 9.* First, by Proposition 8, we know that $a_t$ satisfies the following differential inequality:

$$|\dot{a}_t| \leq C a_t^3 e^{-\frac{\mu_{GN}}{2} t},$$

with $C := \frac{L}{\mu_H \mu} \Delta_0 C_R$. We will control $a_t$ by application of Grönwall's inequality. To this end, consider the ODE:

$$\dot{b}_t = C b_t^3 e^{-\frac{\mu_{GN}}{2} t}, \qquad b_0 = a_0 > 0.$$

We know by Lemma 5 in Appendix B.5 that $b_t$ is given by:

$$\begin{aligned}
b_t &= b_0 \left( 1 + 4C \mu_{GN}^{-1} b_0^2 \left( e^{-\frac{\mu_{GN}}{2} t} - 1 \right) \right)^{-\frac{1}{2}}, \\
&= a_0 \left( 1 + \zeta 2 C_R \Delta_0 \mu_{GN}^{-1} a_0^2 \left( e^{-\frac{\mu_{GN}}{2} t} - 1 \right) \right)^{-\frac{1}{2}}
\end{aligned}$$

for all times $t < T^+$. Therefore, by Grönwall's inequality, it must hold that $a_t \leq b_t$ for all times $t < \min(T_R, T^+)$. Moreover, recalling now that $\|\dot{w}_t\| \leq L \mu^{-1} \mu_H^{-1} \Delta_0 a_t e^{-\frac{\mu_{GN}}{2} t}$ by Proposition 8, we directly get that:

$$\|\dot{w}_t\| \leq \frac{C}{C_R} a_0 e^{-\frac{\mu_{GN}}{2} t} \left( 1 + 4C \mu_{GN}^{-1} b_0^2 \left( e^{-\frac{\mu_{GN}}{2} t} - 1 \right) \right)^{-\frac{1}{2}}. \tag{20}$$

Integrating the above inequality in time yields the following estimate on $\|w_t - w_0\|$ for any $t < \min(T_R, T^+)$:

$$\|w_t - w_0\| \leq \int_0^t \|\dot{w}_s\| \, \mathrm{d}s$$

$$\leq \frac{1 - \left(1 + 4C\mu_{GN}^{-1} a_0^2 \left(e^{-\frac{\mu_{GN}}{2}t} - 1\right)\right)^{\frac{1}{2}}}{C_R a_0}$$

$$\leq 4L(\mu\mu_H\mu_{GN})^{-1}\Delta_0 a_0 = \zeta\Delta_0 a_0,$$

where the second line follows by explicitly integrating the r.h.s. of (20) while the last line follows by the concavity of the square-root function. $\qquad\square$

Proposition 9 shows that the Frobenius norm of the $J_t^\dagger$ remains bounded at all times provided that $\Delta_0 a_0^2$ is small enough, a quantity that depends only on the initial conditions. Moreover, it also shows that making the product $\Delta_0 a_0$ small enough at initialization ensures that $w_t$ remains in a ball of radius $R$ around $w_0$. We exploit these two consequences to prove Proposition 3 in Appendix B.4.

## B.4   Absence of blow-up for almost-optimal initial linear layer: proof of Proposition 3

To prove Proposition 3, we rely on the results of Appendix B.3 which allow controlling the evolution of the smallest singular value of the NTK matrix $A_{w_t}$. More precisely, Proposition 3 is a direct consequence of Proposition 9 with a particular choice for the initialization $w_0$.

*Proof of Proposition 3 .* By assumption on $w_0$, it holds that

$$\Delta_0 := \|\nabla\mathcal{L}(f_{w_0})\| < \epsilon = \frac{\mu\mu_H\mu_{GN}}{8LN}\min(R, C_R^{-1})\min\left(\sigma_0, \sigma_0^2\right).$$

Moreover, by definition of $a_0 := \left\|J_0^\dagger\right\|_F$ and of $\sigma_0^2 = \sigma_{min}(G(u_0))$, we know that $a_0^2 < N\sigma_0^{-2}$. Hence, we get

$$\Delta_0 < \frac{1}{2\zeta}\min(R, C_R^{-1})\min\left(a_0^{-1}N^{-\frac{1}{2}}, a_0^{-2}\right),$$

where we introduced $\zeta := \frac{4L}{\mu\mu_H\mu_{GN}}$. This above inequality directly yields:

$$\zeta\Delta_0 a_0 \leq \frac{R}{2N^{\frac{1}{2}}}, \qquad \zeta C_R\Delta_0 a_0^2 < 1, \tag{21}$$

Therefore, we can use the above inequalities in the estimates (19) of Proposition 9 to get:

$$\|w_t - w_0\| \leq \frac{R}{2N^{\frac{1}{2}}}, \qquad a_t \leq a_0\left(1 - \zeta C_R\Delta_0 a_0^2\right)^{-\frac{1}{2}},$$

for any time $t \in [0, T_R)$, where $T_R$ is defined as:

$$T_R := \sup\left\{t \geq 0 \quad | \quad \|w_t - w_0\| < R, \text{ and } \sigma^\star(A_{w_t}) > 0\right\}.$$

This implies, in particular, that $w_t$ never escapes the ball $\mathcal{B}(w_0, R)$. Moreover, using (16) we get that the smallest singular value $\sigma^\star(A_{w_t}^\top)$ is always lower-bounded by a positive constant $\tilde{\sigma} := a_0^{-2}\left(1 - \zeta C_R\Delta_0 a_0^2\right)$. Therefore, $T_R$ must necessarily be greater or equal to $T$, the maximal time over which $w_t$ is defined. Additionally, if $T$ was finite, then $w_t$ must escape any bounded domain. This contradicts the fact that $w_t \in \mathcal{B}(w_0, R)$. Therefore, $T$ must be infinite and the inequality (6) applies at all times $t \geq 0$. $\qquad\square$

## B.5   Auxiliary results

**Lemma 1.** *Let $J$ be a linear operator from a Hilbert space $\mathcal{W}$ to a finite-dimensional space $\mathcal{H}$ and assume that $JJ^\top$ is invertible. Define the projector $P = J^\top(JJ^\top)^{-1}J$. Then the following relations hold:*

$$(J^\dagger)^\top(I - P) = 0$$

**Lemma 2.** *Let $J$ be a linear operator from a Hilbert space $\mathcal{W}$ to a finite-dimensional space $\mathcal{H}$ and assume that $JJ^\top$ is invertible. Then for any non-negative number $\epsilon$, it holds that:*

$$\left\| JJ^\top (JJ^\top + \epsilon I)^{-2} \right\|_F \leq \left\| J^\dagger \right\|_F^2$$

**Lemma 3.** *Let $\mathcal{W}$ be a Hilbert space and $\mathcal{R}$ be a finite-dimensional euclidean space. Let $J$ be a linear operator from $\mathcal{W}$ to $\mathcal{R}$ and $H$ an invertible operator from $\mathcal{R}$ to itself. Further, assume that $JJ^\top$ is invertible. Then the following holds for any non-negative number $\epsilon$:*

$$\left\| H^{-\frac{1}{2}} \left( JJ^\top + \epsilon H^{-1} \right)^{-1} H^{-\frac{1}{2}} \right\|_{op} \leq \frac{1}{\epsilon + \sigma^\star(H)\sigma^\star(JJ^\top)}.$$

**Lemma 4.** *Let $J$ and $K$ be two Hilbert-Schmidt operators between two Hilbert spaces. Then, it holds that:*

$$\left\| JK^\top \right\|_F \leq \|J\|_{op}\|K\|_F \leq \|J\|_F\|K\|_F$$

**Lemma 5.** *Let $C$ and $r$ be two positive constants. For a given initial condition $b_0 > 0$, consider the following differential equation:*

$$\dot{b}_t = Cb_t^3 e^{-rt}.$$

*Then $b_t$ is defined for any time $t < T^+$ where $T^+$ is given by:*

$$T^+ = \begin{cases} r^{-1}\log\left(1 - \frac{r}{2Cb_0^2}\right), & 2Cr^{-1}b_0^2 > 0 \\ +\infty & \textit{Otherwise.} \end{cases}$$

*Moreover, $b_t$ is given by:*

$$b_t = b_0\left(1 + 2Cr^{-1}b_0^2\left(e^{-rt} - 1\right)\right)^{-\frac{1}{2}}, \qquad t < T^+.$$

*Proof.* First note that a local solution exists by the Cauchy-Lipschitz theorem. Moreover, it must never vanish since otherwise, it would coincide with the null solution $b_t = 0$ by uniqueness. However, this is impossible, since $b_0 > 0$. We can therefore divide the ODE by $b_t^3$ and explicitly integrate it which gives:

$$b_t^{-2} = b_0^{-2} + 2r^{-1}C\left(e^{-rt} - 1\right).$$

The solution is defined only for times $t$ so that the r.h.s. is positive, which is exactly equivalent to having $t < T^+$. □

## C  Additional experimental results

### C.1  Multi-seed experiments

We present result obtained for 5 independent runs. Each run uses a different initialization for the parameters of both student and teacher networks. Additionally, the training and test data are all generated independently for each run. All these results are obtained using `SiLU` non-linearity. It is clear from Figure 3, that the results display little variability w.r.t. the seed of the experiment, which justifies the single-seed setting chosen in the main paper.

**Effect of over-parameterization**  Figure 4 (Left) shows the effect of over-parameterization of the network on generalization error. It appears that the generalization error remains stable with an increasing parameterization for both GN and GD, as soon as the network has enough over-parameterization to fit the data exactly. On the other hand, the test error of RF improves with increasing over-parameterization which is consistent with [22].

**Evolution of the smallest singular value of $A_w$.**  Figure 4 (Right) shows that the smallest singular value $\sigma^\star(A_w)$ systematically increases during training, especially in the feature learning regime (bright red colors). Hence, the dynamics are far from blowing up even as the features change substantially. On the other hand, $\sigma^\star(A_w)$ remains nearly constant for large values of $\tau_0$ (darker colors) which indicates that the features barely change in the kernel regime.

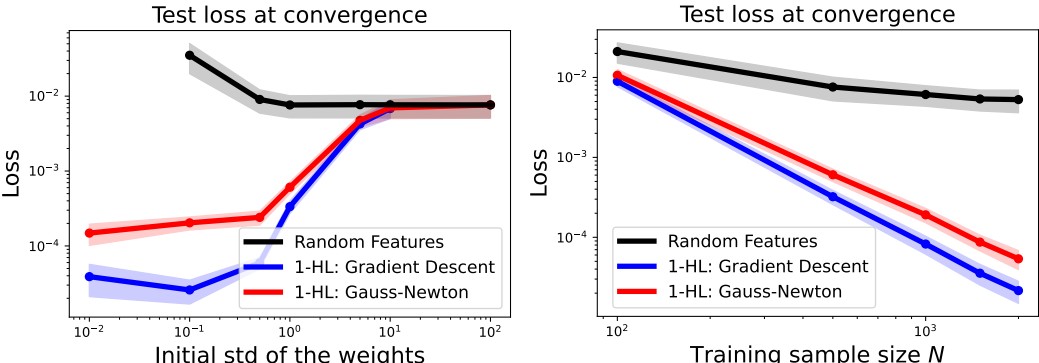

Figure 3: (Left) Test error vs std of the initial weights $\tau_0$. All models are trained until the training objective is smaller than $10^{-6}$ using $M = 5000$ hidden units and $N = 500$. Confidence interval at $95\%$ estimated from 5 independent runs shown in shaded colors. (Right) Test error vs training sample size. All models are trained until the training objective is smaller than $10^{-6}$ using $M = 5000$ hidden units and initial std of the weights $\tau_0 = 1$. Confidence interval at $95\%$ estimated from 5 independent runs shown in shaded colors.

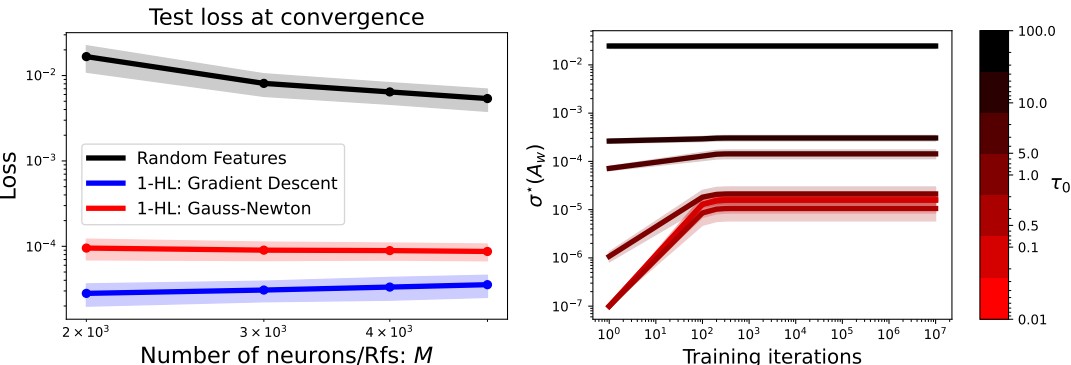

Figure 4: (Left) Test error vs number of hidden units, All models are trained using $N = 2000$ training samples until the training objective is smaller than $10^{-6}$. The initial std of the weights is set to $\tau_0 = 1$. (Right) Evolution of $\sigma^\star(A_w)$ during training using Gauss-Newton for different values of the initial std $\tau_0$ of the weights. All models are trained using $N = 500$ training samples until the training objective is smaller than $10^{-6}$. We used $M = 5000$ units. Confidence interval at $95\%$ estimated from 5 independent runs shown in shaded colors.

## C.2 Effect of the sample-size

Figure 5 shows the evolution of the test error as the training sample size $N$ varies in $\{100, 200, 500, 1000, 1500\}$ for two choices of activation functions: `ReLU` (Left) and `SiLU` (Right). In both cases, we display the results for two different values for the initial std $\tau_0$ ($10^{-3}$ and $1$). Results are reported for the best performing step size for each of GN and GD. The first observation is that the performance gap between random features RF and learned features (using either GN or GD) keeps increasing with the sample size. This observation suggests that the learned features use training samples more efficiently than random features. The second observation is the almost affine relation between the generalization error $\mathcal{L}_{test}$ and training sample size in a logarithmic scale with a strong dependence of the slope on the variance at initialization and the optimization method. Such affine relation implies the following upper-bound on $\mathcal{L}_{test}$ in terms of sample-size $N$:

$$\mathcal{L}_{test} \leq C \frac{1}{N^\alpha},$$

for some positive constants $C$ and $\alpha$. The coefficient $\alpha$ controls the speed at which the estimator approximates $f^\star$ as one accesses more training data and usually appears in learning theory to describe

the statistical convergence rate of a given estimator [46]. The consistent and strong dependence of $\alpha$ on the optimization methods (GN vs GD) and regime (kernel vs feature learning) further confirms the implicit bias of both initialization and optimization algorithms.

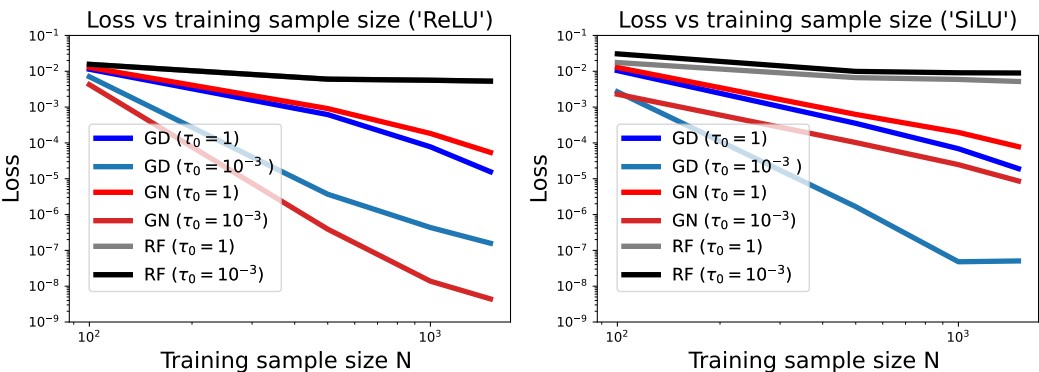

Figure 5: Final values of various metrics vs training sample size for both ReLU (left) and SiLU (right) networks. Two values for $\tau_0$ are displayed for each method $\tau_0 \in \{10^{-3}, 1\}$ while training data ranges from $N = 100$ to $N = 1500$. All models are optimized until the training objective is smaller than $10^{-6}$ using $M = 5000$ hidden units. For both GD and GN, results are reported for the best-performing step-size $\lambda$ selected according to the test loss on a regular logarithmic grid ranging from $10^{-3}$ to $10^3$.

## C.3 Choice of the non-linearity

Figure 6 shows the effect of step-size on the final performance of a SiLU network with two choices for the temperature parameter: $\beta = 1$ and $\beta = 10^6$. Similarly to Figure 1, it is clear that increasing the step size in Gauss-Newton results in features that do not generalize well, while the opposite is observed in case of gradient descent. On the other hand, depending on the value of $\beta$, we observe that GN either outperforms GD ($\beta = 10^6$) or the opposite ($\beta = 1$). This variability suggests that the effectiveness of an optimization method in improving generalization is dependent on the specific characteristics of the problem at hand. Finally, note that for $\beta = 10^6$, we recover almost the same results as those obtained for ReLU in Figure 1 (right). This is due to the fact SiLU becomes a tight smooth approximation to ReLU for large values of $\beta$. We also notice that when using a larger std for the initial weights (ex. $\tau_0 = 1$), using either ReLU or SiLU with ($\beta = 1$) yields similar results, as shown in Figure 7.

## C.4 Experiments on MNIST

We have performed a series of experiments on MNIST to illustrate the behavior of GN on higher dimensional data (notably, with multidimensional targets). To stay as close as possible to the theoretical framework, we modify the original MNIST dataset. Specifically, we construct a student/teacher setup in which the trained network can achieve a zero-loss, which is an assumption made in Section 3.1.

**Building datasets based on MNIST by using a teacher.** In our theoretical framework, we restricted ourselves to cases where the minimum loss can be achieved on the considered model. To make this condition hold, we build a new dataset by using a teacher trained on the MNIST classification task [16]:

1. we construct a training set for the teacher which consists of $\mathcal{D}_{tr} \cup \mathcal{D}_{ts}$, where $\mathcal{D}_{ts}$ is the original MNIST test dataset and $\mathcal{D}_{tr}$ is a balanced subset of size 3000 of the original MNIST training set;

2. we then train a teacher neural network with one hidden layer of size 50 and ReLU activation function on training set $\mathcal{D}_{tr} \cup \mathcal{D}_{ts}$ using a cross-entropy objective;

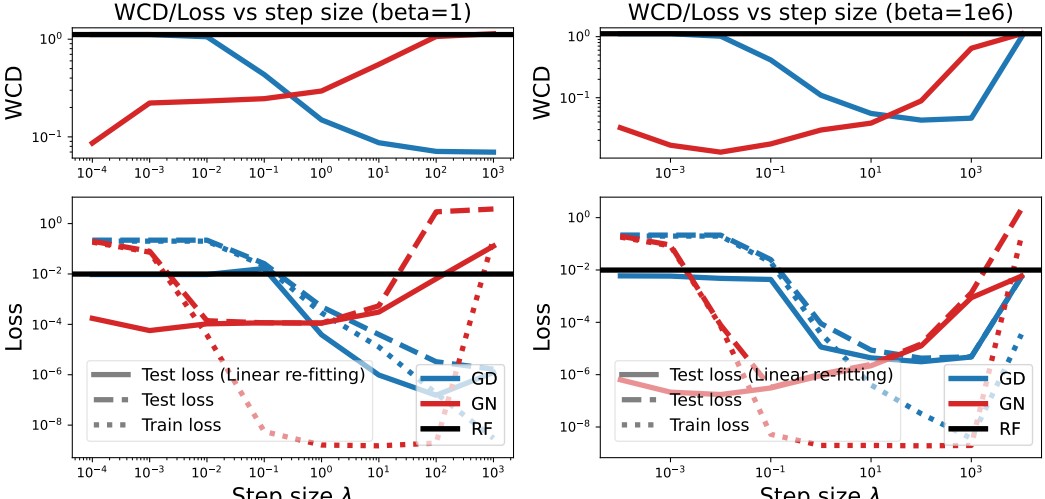

Figure 6: Final values of various metrics vs the step size for both SiLU network with $\beta = 1$ (left) and $\beta = 10^6$ (right). (Right figure) The std of the weights at initialization is set to $\tau_0 = 10^{-3}$. All models are optimized up to a training error of $10^{-6}$ or until the maximum number of steps is exceeded, ($M = 5000$, $N = 500$).

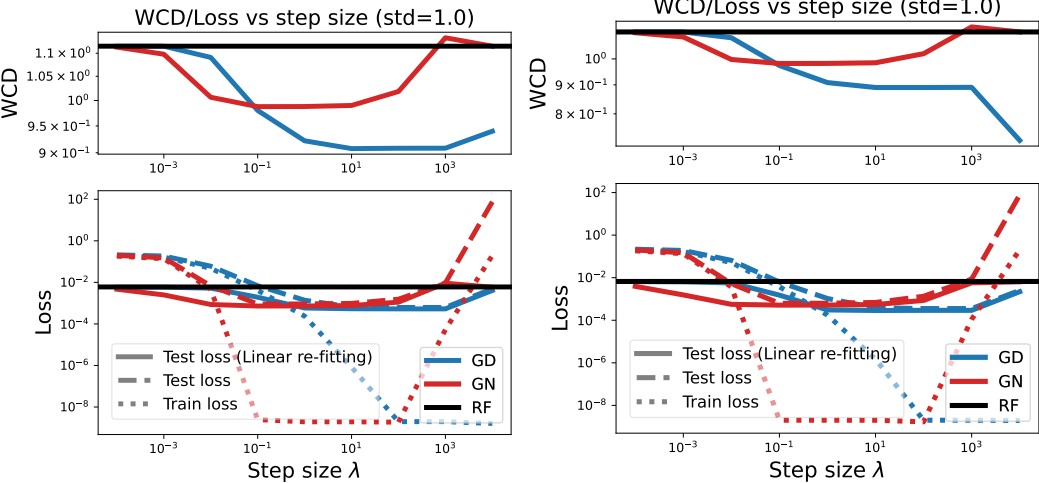

Figure 7: Final values of various metrics vs the step size for both ReLU (left) and SiLU (right) networks. The std of the weights at initialization is set to $\tau_0 = 1$. All models are optimized up to a training error of $10^{-6}$ or until the maximum number of steps is exceeded, ($M = 5000$, $N = 500$).

3. given the trained teacher $f_T$, we build new datasets $\bar{\mathcal{D}}_{tr}$ and $\bar{\mathcal{D}}_{ts}$ for the student network:

$$\bar{\mathcal{D}}_{tr} := \{(x_i, \text{Softmax}(f_T(x_i))) : (x_i, y_i) \in \mathcal{D}_{tr}\},$$
$$\bar{\mathcal{D}}_{ts} := \{(x_i, \text{Softmax}(f_T(x_i))) : (x_i, y_i) \in \mathcal{D}_{ts}\}.$$

**Training procedure.** Once the pair of training and testing datasets $(\bar{\mathcal{D}}_{tr}, \bar{\mathcal{D}}_{ts})$ has been built, we use them to train and test a *student* neural network based on (10), with one hidden layer of size 5000 and a ReLU activation function. We use a similar initialization scheme for the student's network and fix the std of the initial weights to $\tau_0 = 0.001$. For training, we use a quadratic loss between the student and teacher's outputs and perform a maximum of 100000 iterations of either GN or GD. The training stops whenever the training loss goes below $10^{-6}$. We vary the step size $\lambda$ on a logarithmic range $[10^{-3}, 10^3]$ for both methods and evaluate the test loss and test loss after linear re-fitting.

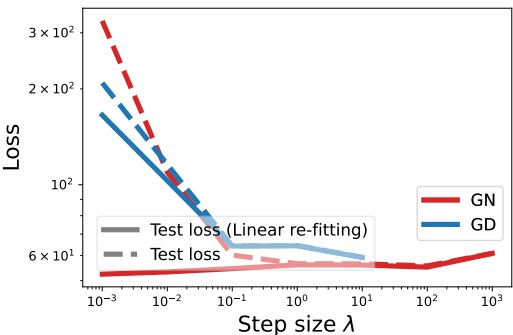

Figure 8: Final values of various metrics vs the step size for a ReLU network trained using MNIST data. Std of the initial weights is set to $\tau_0 = 0.001$. For GD, results are displayed up to $\lambda = 10$ the algorithm diverges for higher values.

**Results.**    Figure 8 shows the evolution of the test loss (both with or without re-fitting) as a function of the step size. We observe similar behavior as in the main experiments, with hidden learning occurring for GN when using small step sizes and a degraded generalization error as the step size increases. On the other hand, generalization improves for GD by increasing the step size up to the point where optimization becomes unstable.

