# OpenReview forum: "Rethinking Gauss-Newton for learning over-parameterized models"
_NeurIPS.cc/2023/Conference — NeurIPS 2023 poster_

### Official Review · Reviewer_oHC4 · 2023-06-10

**Soundness:** 3 good
**Presentation:** 3 good
**Contribution:** 1 poor
**Rating:** 3
**Confidence:** 4

**Summary:**

This paper studies applying the Gauss-Newton method to train the neural network to solve a regression problem in the NTK and mean-field regime. The paper establishes a linear convergence rate on Gauss-Newton in this setting. The authors further provide experiments in the teacher-student network setting to study the effect of initialization and step size on convergence and generalization.

**Strengths:**

The presentation of this paper is quite good. All the proofs and theorems and experimental results are well-documented. The proofs and experiment results in the appendix are very nicely structured and organized, making it very clear for the reviewer to understand what the authors are doing.

**Weaknesses:**

The major weaknesses of this work is on its contribution. First of all, the Gauss-Newton method is a well-known classical method in convex optimization to achieve a faster super-linear convergence rate than first-order methods such as gradient descent which can only achieve linear convergence rate. Therefore, it is natural to apply this method to improve the optimization on training neural network. However, the challenging problem in training neural network is due to its non-convexity. The authors consider the NTK regime and mean-field regime. It is well-known that the optimization problem in the NTK regime is strongly-convex-like. Therefore, the results the authors established in the paper is non-surprising and further, proposition 1 is only able to establish a linear convergence rate whereas Gauss-Newton is able to achieve locally super-linear convergence rate in the classical convex setting, with some additional assumptions needed to be made. Note that if Gauss-Newton is only able to achieve linear convergence, then the advantage of Gauss-Newton is non-existent over gradient descent since gradient descent can also converge linearly and the cost per-step is much smaller than Gauss-Newton. Also, the goal of optimization in deep learning is for the model to generalize, and it is not clear to me how the Gauss-Newton method can affect the model's generalization.

Therefore, currently, I am issuing a rejection for this submission.

**Questions:**

Due to the current scale of NeurIPS and the workload on the reviewer, I am unable to go through the appendix in detail. The authors claim their results also holds for the mean-field regime and the proofs in the appendix seems to only address the NTK regime. I encourage the authors to clarify their results. I think for the mean-field regime the authors may have made some assumptions like the initial point is close to the true minimizer which can circumvent the problem of dealing with non-convexity in the mean-field regime. However, this, again, returns back to the classical convex regime.

**Limitations:**

The authors discussed the limitations in the conclusion section.

---

> ### Author Rebuttal · Authors · 2023-08-09
>
> Thank you for your review. We believe the following clarifications should address all your concerns about the strength of our contribution. If you still have concerns, we are more than happy if you could kindly share them as this would help us improve this work.
>
> 1. **Convergence:** The non-convexity and degeneracy of the loss introduces challenges do not appear when studying GN for strongly convex objectives. This is apparent in the proofs which do not rely on classical convexity arguments. Below, we discuss the relevance of the result in terms of linear vs super-linear rate and mean-field limit.
>    - “Linear vs super-linear rate”: The linear rates we provide are general as they include both GN (when the matrix H is the hessian) and natural gradient (when H is the identity). The super-linear rates are only possible when choosing H to be the hessian and are only local. Moreover, Natural gradients are still useful and extensively studied despite their linear rate (see next point).
> Improved conditioning: the linear rate for GN still has much better conditioning than gradient descent which justifies its advantage over gradient descent even if the cost per iteration is higher. Our result highlights precisely this.
>
>    - Mean-field limit: Indeed, the convergence result assumes the initial point is close to optimality. This is stated in Proposition 2 of the main text: L215-216. By mean-field limit, we mean using a scaling of $1/M$ instead of $1/\sqrt(M)$ where $M$ is the number of particles (L140). Both NTK and mean-field are captured in proposition 2. That being said, we clarify that this does not recover the classical convex regime for two reasons:
>       - The loss is not convex even near the local minimizer: this was shown in Liu et al  2021. Hence, one can’t use convexity arguments in parameter space.
>       - The pre-conditioner in GN can become singular and make the dynamical system diverge (also called finite-time blow-up). This never happens in the strongly convex case where the pre-conditioner has a lower-bounded singular value. This is precisely what makes the analysis more challenging than the strongly convex case.
>
> 2. **Generalization of GN:** These are precisely the empirical contributions of the paper:
>    - GN generalizes better with smaller step sizes and exhibits a phenomenon of “hidden learning”: Both train and test losses can remain large when using small step sizes because the linear layer is not well fitted, still the hidden features keep improving. This ultimately yields better generalization after fitting the linear layer alone (Figures 1 and 2).
>
>    - We show that using a larger step size for GN does not generalize well despite being faster at optimizing the training objective: This challenges the common belief of using a larger step size for GN.
>
>    - We show empirically that GN can generalize well, even better than GD. This challenges a common belief that GN does not find solutions that generalize well.

---

> > ### Author Response · Authors · 2023-08-14
> > **We would be grateful if you engage with us during the discussion period**
> >
> > Dear reviewer,
> >
> > We thank you for your time.
> >
> > Since the authors/reviewers discussion period is still ongoing, we would be grateful if you give us the opportunity to engage in a constructive discussion about our submission. We are grateful that two reviewers increased their scores based on the constructive discussions we had with them, including c9sb who now recommends an accept with a score of 7. We hope you can give us a chance to clarify any concerns you might have.
> >
> >
> > Our contributions are twofold:
> >    - A theoretical result on global convergence of GN
> >    - An empirical study of the generalisation of GN
> >
> > From your review, we understand you had concerns only about the global convergence result which we believe we addressed as follow:
> >    - **Strength of the result: Linear rate instead of super-linear:** Our result provides a global rate and also covers natural gradients which can only have a linear rate. Super linear rates hold only locally.
> >    - **Advantage of GN/natural gradient with a linear rate compared to GD:** The advantage comes from an **improved conditioning** compared to GD. Note that natural gradient methods can only come with a linear rate, have a similar cost pre iteration as GN and yet are of interest. Our result covers natural gradients as well.
> >    - **Novelty of the result: Mean-field vs NTK regime** Obtaining global convergence results in the mean-field limit is still an active area of research, even for gradient descent ([10,12]) and does not rely on strongly-convex-like arguments as in the NTK regime. This work is the first to provide a convergence result for GN in the mean-field limit.
> >
> > From these clarifications, do you still have other doubts based on which you recommend rejection?
> >
> > Thanks again for your time.

---

> > ### Comment · Reviewer_oHC4 · 2023-08-16
> > **Thank you for your response.**
> >
> > I thank the authors for the clarification.
> >
> > I would like to clarify that in my original review, when I was saying "strongly-convex-like", I am referring to the case that you can lower bound the smallest eigenvalue of the NTK. *Although the objective is non-convex, the analysis is essentially a convex analysis since you are assuming the initial point is close to the global optimal (in proposition 2) which can help you avoid dealing with the non-convexity by utilizing the positive smallest eigenvalue of the NTK.* With such an assumption, the convergence analysis of NTK and mean-field are essentially the same. There are further issues like in Equation 7, you are assuming the damping parameter is chosen to be some positive constant multiplied the smallest eigenvalue of the NTK. Although this later help you prove convergence, why this assumption is practical? Estimating the smallest eigenvalue of the NTK is costly in practice, let alone obtaining the exact value. Therefore, for now, I am keeping my original evaluation.

---

> > > ### Author Response · Authors · 2023-08-17
> > > **On the technical contribution**
> > >
> > > Dear reviewer,
> > >
> > > We thank you for your comment. Below, we provide clarifications about the damping being optional in our results and about the technical challenge addressed in our proofs and that arise from the mean-field regime.
> > >
> > > We would be grateful if you could let us know if you still have points that require more clarifications.
> > >
> > > 1. **The main technical challenge in the proof is to control the smallest singular value in the mean-field regime.** We agree with you that, once a lower-bound on the smallest singular value is assumed to hold, the convergence rate follows easily as shown in the proof proposition 1. However, the challenge is to control the smallest singular values which can get arbitrarily close to 0 in the mean-field regime. We achieved that through three technical propositions stated in the appendix (prop 5, 6 and 7) which are then used to prove the result in proposition 2. We summarize the difference between NTK and mean-field regime captured by these results:
> > >    - **In the NTK regime**, as $M\rightarrow +\infty$, the NTK matrix becomes constant during the optimization dynamics. Therefore, the smallest positive singular value also reamains positive and constant.
> > >     - **In the mean-field regime**, the NTK matrix is not constant during the training dynamics. Even worse, its smallest positive singular value can become arbitrarily close to 0 in general. This happens, for instance, when the variance of the initial weights gets smaller and smaller. Our contribution is to analyse the dynamics of the smallest singular value during optimization and find conditions on the initial weights to prevent it from vanishing thus ensuring convergence.
> > > If the reviewer has a reference in mind where such dynamics of the singular values is studied and controlled in the mean-field regime for Gauss-Newton, we are ready to revise our statement of the contributions.
> > >
> > > 2. **"You are assuming the initial point is close to the global optimal "**:
> > >    - Our assumption does not require the initial point to be within a ball of any global optimum (there is a whole manifold of global optima as we detail next). As stated in proposition 2, it consists in randomly selecting the hidden layer and tuning the linear layer only until the functional gradient is below a pre-determined threshold. Note that this condition allows the hidden weights to evolve in an arbitrarily large radius $R$ within the initial ones provided the linear layer provides a good enough fit (as measured by the norm of the functional gradient).
> > >    - A condition of a similar nature has been used in Chizat 2019 [13], Theorem 3.3 and Corollary 3.4 to establish global convergence of the the Wasserstein gradient flow in the over-parameterized regime of two-layer networks in the mean-field limit. We discussed this in L237-241, but will make it clearer.
> > >    - This condition is easy to achieve in practice, since optimizing the linear layer only is guaranteed to minimize the functional gradient norm arbitrarily well.
> > >
> > > 3. **"the analysis is essentially a convex analysis"**: As shown in proposition 2 of [29], the loss landscape of over-parameterized neural networks is non-convex even locally near global optima. But we agree that controlling the smallest positive singular value of the NTK matrix is a key step and in fact the most challenging part in our result.
> > >
> > > 4. **Damping parameter is optional: it can be equal to 0**:
> > >    - **Theory:** Please note that the parameter $\alpha$ is assumed to be non-negative only (can be equal to 0, see L180). Our proofs do not rely on such damping at all. We will make it clear in the text.
> > >    - **Experiments:** We observed little difference when either using damping or not, i.e. setting alpha= 0 or 1 did not change the overall conclusions. That is because the smallest singular value was very small in practice.

---

> > > > ### Comment · Reviewer_oHC4 · 2023-08-18
> > > > **Thank you for your response.**
> > > >
> > > > I thank the authors for the further clarification.
> > > >
> > > > First of all, I want to point out that I got the impression that proposition 2 made an assumption on the initial point being close to the optimality is because the authors mentioned in their rebuttal that ```Mean-field limit: Indeed, the convergence result assumes the initial point is close to optimality.``` I checked the manuscript and the authors indeed didn't make such an assumption. Instead, the result in proposition 2 requires the linear layer to be near-optimal satisfying $\|\nabla L(f_{w_0})\| < \epsilon$. I don't think such assumption can be satisfied by random initialization and as the authors point out it requires pre-training the linear layer so that the gradient is below a certain threshold. Thus, the authors are arguing for a two-stage training algorithm where we first fine tune the linear layer and then use GN to train the hidden layer, instead of plain GN from the very beginning. However, the threshold for terminating the linear layer pre-training requires $R, C_R$, and proposition 2 says $R$ can be any positive number. It is not clear to me why knowing $R, C_R$ is a practical assumption.

---

> > > > > ### Author Response · Authors · 2023-08-18
> > > > > **Thank you for your feedback**
> > > > >
> > > > > Dear reviewer,
> > > > >
> > > > > We are grateful for your feedback and thank you for engaging with us in this discussion which we hope will address all your remaining concerns.
> > > > > We also apologize for the confusion due to our phrasing in the rebuttal on the initial condition and we are grateful that you took the time to double check this in the manuscript.
> > > > > We provide the following clarifications.
> > > > >
> > > > > 1. **Random initialization is not enough for ensuring the linear layer is near-optimal**: Indeed, we completely agree with you on this point. The random initialization is only for the hidden layer $u_0$ . The linear layer $v_0$ needs to be further optimised while maintaining the hidden weight $u_0$ fixed.
> > > > > 2. **Two-stage training:** Indeed our result holds under the condition that we perform an initial fitting step on the linear weights until a threshold $\epsilon$ is reached. Note that a similar type of assumption was used in Chizat 2019 [13], Theorem 3.3 and Corollary 3.4 to establish global convergence of the Wasserstein gradient flow.
> > > > >
> > > > > 3. **Why randomly initialise the hidden weights $u_0$?**
> > > > >    - The random gaussian initialization $u_0$ of the hidden layer allows us to obtain an invertible Gramm matrix formed by the hidden features (equation 4). In other words, the random initialization ensures assumption (A) is satisfied. In section 3.1 (L150-162) we provide references showing that such a matrix is indeed invertible with high probability when the number of parameters is larger than training data. When this matrix is invertible. We can show that a global optimum can be reached by only optimising the linear layer.
> > > > >    - The condition (A) ensures there exists a parameter value $v_0$  for the linear weights achieving 0 training error when using the hidden weights $u_0$. Indeed, under this condition, one can show that, any vector v_0 that is a critical point for the loss (always exists for u_0 fixed) (i.e: $\partial_v L(f_{(v_0,u_0)}) =0$), must be a global minimizer.
> > > > > 4. **Choice of the threshold:** As you point out the threshold depends on two quantities: R and C_R:
> > > > >    - The radius $R$ of a ball B centred around initialization: It doesn’t need to be known, it can be chosen to be any positive number
> > > > >    - Once $R$ is fixed, C_R is the supremum over the B of radius R of the operator norm of the derivative of the Jacobian of the network. In fact using an upper-bound on C_R instead of the supremum works as well in our result (it just makes $\epsilon$ smaller). Because the network is a 1-hidden layer network, finding such upper-bounds can be done in practice.

---

### Official Review · Reviewer_2qbp · 2023-07-06

**Soundness:** 3 good
**Presentation:** 3 good
**Contribution:** 1 poor
**Rating:** 4
**Confidence:** 4

**Summary:**

This work theoretically and empirically studied learning dynamics and generalization properties of one-hidden-layer networks trained with the Gauss-Newton algorithm. The main theoretical result is that the authors provided a loss convergence guarantee, which is provably faster than the same networks trained with gradient descent. The authors then empirically studied generalization properties of the GN-trained NN in a student-teacher setting.

**Strengths:**

The theoretical treatment of the GN in Sec. 4 is well appreciated. While I did not check the correctness of the proof, it appears intuitive that GN methods have better convergence.

**Weaknesses:**

My primary concern is the relevance of the work. My impression is that GN method is virtually never used in NN training, and a brief literature search turned up only a few papers discussing this. This is simply due to the dramatically higher compute demand compared to GD, as the authors mentioned. The size of the Jacobian is N by N, where N is the number of trainable parameters. So training any modern NN with GN can be prohibitively expensive, since it involves pseudo-inversing the Jacobian at every step. While the author mentions other methods such as natural gradient descent, which are in spirit connected to GN, it is unclear how the theoretical/empirical results in this work generalize to those methods, which may be more relevant to this conference.

My second concern, which I am less confident of, is that the fact that GN converges faster than GD (at least in terms of the big O analysis) comes as no surprise. That GN converges faster than GD in a nonlinear least-squared problem appears to be well known (https://www.princeton.edu/~aaa/Public/Teaching/ORF363_COS323/F14/ORF363_COS323_F14_Lec9.pdf). I understand that additional technical requirements need to be proven in the one-hidden-layer NN case, but I'm not sure that they are of importance conceptually.

Finally, the empirical part of the paper and the theoretical paper part are pretty unrelated, besides the fact that they both involve one-hidden layer networks.

**Questions:**

1. How can we generalize the results in this work, theoretically or empirically, to approximate second order methods (e.g. natural GD) that people use to train NNs?

**Limitations:**

See weaknesses.

---

> ### Author Rebuttal · Authors · 2023-08-09
>
> Thank you for your review. We believe there is a misunderstanding of the relevance of our contribution. The following clarifications should address all your concerns. If you still have other concerns, we are more than happy if you could kindly share them as this would help us improve this work.
>
> 1. **Relevance:**  We believe there is a big misunderstanding here:
>    - **Natural gradient is covered in our work as discussed in L176.**  It corresponds to taking the matrix H to be the identity as discussed in L176. The result of proposition 2 holds for this case as well.  The experimental section corresponds also to a natural gradient on a Gaussian model where the network represents the meanwhile the variance is fixed. In that particular case, GN and natural gradient have the same updates (see ref [31] ). Hence, our work shows that GN and Natural gradient can find solutions that generalize well. We believe this to be highly relevant to the community interested in second-order methods for deep learning.
>
>    - **The computational cost:** Does not require inverting a matrix of the size of the parameters**, only the size of a batch of data. This is thanks to the Woodbury matrix identity that is commonly used for these methods. We use it on a full batch of data and discuss it in L282-284 (note that in the paper N denotes the sample size not the number of parameters). Using smaller batches is scalable and was shown to work well in practice (see ref 11). In our setting, we use a full batch as we focus on the ideal algorithm GN without introducing additional biases or stochasticity in the estimation which can interfere with the interpretation of the results. This approach allows us to understand the ideal performance that a scalable approximation of GN can aim towards.
>
>
>
> 2. **Novelty/significance of the convergence result:**:
>    - The convergence result you mentioned in the link holds for invertible Hessians and is only local. This is not applicable to over-parameterized networks where the hessian is degenerate by construction. This makes the dynamical system prone to diverge in finite time (blow-up) and is more challenging to deal with. Our result precisely handles this possible blow-up (see section A.2 of the appendix) using non-standard techniques that go beyond convex analysis and that require dynamical systems analysis instead.
>    - **“I'm not sure that they are of importance conceptually”:** The convergence result of GN might seem intuitive when thinking about the invertible hessian case. However, there is a gap between conjecturing the result from this particular case and rigorously proving it in the more challenging scenario of non-convex over-parameterized networks. Our result relies on non-standard optimization techniques which can still be insightful. In particular, the technical tools used come from dynamical system analysis (as discussed in L225-230) and are different from convex analysis often exploited to establish convergence of GN.
>
> 3. **Theoretical vs empirical parts:** The motivation for using GN/natural gradient is the faster convergence rate for the training objective. However, these are known for convex objectives with non-degenerate hessian. Two questions stand in the way to understand if GN/natural gradient is useful for learning over-parameterized neural networks:
>    - When can GN reach a global solution for the training objective? The theory part addresses the first point and focuses only on the training objective. The empirical part confirms the global convergence since GN systematically obtains 0 training error faster than GD.
>    - Does such a solution generalize well to test data? The empirical part studies the generalization of properties of the solution obtained by GN compared to GD. It provides two practical insights:
>        - The commonly used prescription for GN is to use the largest possible step size to achieve faster convergence of the training objective and benefit from the faster rate of GN/natural gradient. We show that this results in poor generalization (similar to random features without learning the hidden layer).
>        - Instead, we show that smaller step sizes for GN allow better feature learning and are thus desirable but they come at a higher computational cost.

---

> > ### Comment · Reviewer_2qbp · 2023-08-12
> > **Thank you**
> >
> > My sincere thanks for the authors' detailed response. I hereby acknowledge that I have read the rebuttal.
> >
> > I appreciate the clarification of the connection to natural gradient descent as well as the computational cost. This does allay a major concern of mine.
> >
> > In terms of the novelty/contribution, I admit that such judgment is always subjective, but I am still not convinced. In the rebuttal the authors allege that the work is novel because this is in the overparameterized regime. But ref 11 that the authors cited, for example, supplies a convergence analysis in the overparameterized limit. Again, I understand that this work covers a technically different scenario than previous work, but I do not find the conceptual picture different enough.
> >
> > Finally, for the theory-empirical gap, I still think that the experiments are not well motivated by the theory (reviewer c9sb raises a similar concern). I think the experiments could be an interesting research project on its own (an extensive empirical study of the best LR to use for GN for generalization).
> >
> > I have raised my score in response to the rebuttal.

---

> > > ### Author Response · Authors · 2023-08-13
> > >
> > > Dear reviewer,
> > >
> > > Thank you for engaging with us during the discussion period and for remaining open to raising your score as a response to our comments.
> > > We are also glad to read that you find our experimental contribution to be an interesting research project on its own. We would also like to bring to your attention the updated results on MNIST that we provided in response to reviewer c9sb as they further support our paper.
> > > As we understand it, the remaining concerns are about the **theoretical contribution** and the **connection between theory and experiments**, which we will address below. Please let us know if you have additional questions.
> > >
> > > 1. **Novelty of the theoretical result**:
> > >    - **"The authors allege that the work is novel because this is in the overparameterized regime:"** We apologize if we gave the impression that the overparameterized regime was the sole reason we claim our work to be novel, we will make that clearer in the text: The novelty comes from the conjunction of two settings: **Overparameterized setting** and  **mean-field limit**. The paper [11] considers the **overparameterized setting in the NTK limit** since they use the limiting properties of a network with a scaling of $1/\sqrt{M}$ in their analysis. We use a scaling of $1/M$ which yields a different limiting object: mean-field limit. While this might sound only a technical difference, it has important conceptual implications as we detail next.
> > >    - **Technical difference**. Obtaining global convergence results in the mean-field limit is still an active area of research, even for gradient descent ([10,12]). We believe this work to be the first to provide a convergence result for GN in the mean-field limit. This requires a different analysis than the NTK limit. Indeed, since the NTK limit is essentially a kernel method with hidden weights remaining constant during optimization, the model behaves as a linear model and convergence results follow from a strong convexity-like argument. This is not the case for the mean-field limit where the evolving hidden weights result in a highly non-convex objective. We will add a paragraph in the related work section to emphasize this difference.
> > >    - **Conceptual difference:** The mean-field limit is relevant for obtaining good generalisation in neural networks. It is now well documented that the NTK limit is equivalent to a kernel method/linear model. Meaning that only the linear layer is effectively trained, while the hidden layers remain almost close to initialization [1,24] . On the other hand, the mean-field limit allows feature learning, i.e. hidden layers change during optimization to learn a good data representation. This results in better generalisation (see also Figure 5 of the appendix which illustrates the difference in generalisation). The mean-field limit is more consistent with what is observed in practice when training a neural network that generalises well. It is therefore of interest to consider this particular setting.
> > >
> > >
> > > 2. **Theory-empirical gap**. We apologize if the transition between experiments and theory gives the impression of a gap. We will add an explicit discussion between the two parts in the experiment section to emphasize the connection between the two.
> > >    - **Reviewer c9sb acknowledged it was no longer a concern and raised their score to accept**. Please see our response to reviewer c9sb and their new reply.
> > >    - **The experiments are consistent with the theory:** Figure 2 (bottom right) shows that the training objective converges faster when using GN than when using GD as illustrated by Proposition 2. Moreover, in all experiments, GN optimises the training objective globally, as predicted by the proposition. Note that in practice, the global optimum is reached even without the initial condition on the linear weights of proposition 2. This suggests that the condition could be relaxed. Please also refer to the **Pre-training the last layer to near optimality** paragraph in the answer to reviewer c9sb on this matter.
> > >    - **The experiments provide additional insights not covered by the theory:** The experiments illustrate that obtaining a good generalisation requires more than just optimizing globally the training loss. While Proposition 2 provides a global convergence result for the training loss, it says nothing about feature learning and is “happy” with the features staying near initialization as long as the loss reaches a global solution. However, the amount of feature learning during optimization is what essentially drives generalisation performance. Our experiments illustrate this precisely and thus puts the result of proposition 2 into perspective.

---

> > > > ### Author Response · Authors · 2023-08-18
> > > >
> > > > Dear reviewer,
> > > >
> > > > We thank you for your time and engagement with our work.
> > > > We wanted to know your thoughts about the answers we provided to your previous comment. In particular, if there are any remaining concerns that you might have.
> > > >
> > > > Thank you
> > > >
> > > > The authors

---

### Official Review · Reviewer_c9sb · 2023-07-06

**Soundness:** 2 fair
**Presentation:** 4 excellent
**Contribution:** 2 fair
**Rating:** 7
**Confidence:** 3

**Summary:**

This paper investigates theoretically and empirically the implicit biases of the Gauss-Newton (GN) optimization algorithm on over-parametrized one hidden layer-models (e.g. the capacity of the model is much higher than that of the ground-truth function to approximate). The main theoretical contribution of the paper pertains to global optimization:  a global convergence rate is derived when dynamics do not blow up in finite time, and to provide sufficient conditions on the initial state of the model to prevent this blow up. The experimental section explores generalization: it compares GN and Gradient Descent (GD) for various initial hidden weights variances and learning rates. The key take-away of this last section (and empirical prescription thereof) is that smaller learning rates are better for GN (while the converse holds for GD) and that GN under optimizes the linear layer (e.g. opposite of the lazy regime) while learning good hidden features.

More precisely:

- Section 3.1 introduces the hypothesis at use on the ground-truth data, the learning objective (Eq. 1) and the model at use, in the finite width (Eq. 2) and mean-field infinite width (Eq. 3) cases. Overparametrization (A) is defined in terms of the invertibility of the feature kernel.

- Section 3.2 defines the generalized GN vector field (Eq. 5) and defines associated discrete and continuous weight dynamics (Eq. 6). Hypothesis on the parameters coming into play in the GN dynamics are introduced (B and Eq. 5) in order to derive subsequent global convergence results.

- Section 4.1 presents Proposition 1 which states that under the aforementioned assumptions, the GN dynamics either blow up or converge to a global minimizer of the loss with an explicit rate which only depends on specific constants characterizing the functions at play, which is in stark contrast with SGD convergence rate which is controlled by the small singular value of the Neural Tangent Kernel (NTK).

- Section 4.2 presents Proposition 2 which states that it is sufficient, for the dynamics not to blow up and converge that the rate derived previously, that the linear layer is near optimal (e.g. $\\|\nabla_v \mathcal{L}(f(u_0, v_0))\\|$ sufficiently small at initialization).

- Section 5.1 presents the teacher-student experimental setup, where the ground-truth data $X$ is Gaussian (with dimension $d=10$) with associated labels $Y$ generated by a shallow one hidden-layer model (5 hidden units) and the student model is a wide/overparametrized one hidden-layer model (5000), both with a ReLU activation. The three optimization algorithms at use are presented: GN, GD and random features (RF) whereby the hidden weights are random and the linear weights are taken as the minimal-norm least square solution (Eq. 9).

- Section 5.2 presents the performance metrics, i.e. the weighted cosine distance (WCD) between teacher and student features (which allows for teacher and student features of different dimensions, promotes feature alignment and penalizes feature norm) and the test loss after linear re-fitting (Test-LRFit). Test-LRFit is particularly useful to disentangle the analysis of the whole model performance *versus* the quality of the learned hidden features, and subsequent identify when GD and GN operate in the kernel/lazy or feature-learning regime.

- Section 5.3 presents and analyzes training experiments, with varying standard deviations of the visible-to-hidden weights (which we denote here $\sigma_u$ for simplicity) and learning steps. Key results are as follows:
   + Upon increasing values of $\sigma_u$, GD and GN both exhibit a transition between the *feature learning regime* (performance is better than the RF baseline) and the *kernel regime* (performance matches that of the RF baseline).
   + In the feature-learning regime (small $\sigma_u$), GN operates best (in terms of the resulting test loss) with a small learning rate ($\approx 10^{-3}$) with performance degrading onwards while GD operates best as the learning rate increases ($\approx 10^{2}$). GN best performance is better than GD best performance.
   + GN under-optimizes the output linear layer when employing small learning rates while learning good hidden features (as measured by the gap in test loss before and after Linear re-fitting).
   + GN starts learning the internal features (as measured by test loss after linear re-fitting until iteration $\approx 10^4$) and output features onwards (as measured by the test loss). Conversely, last layer and hidden layer are learned simultaneously with GD (i.e. test loss and test loss after linear re-fitting coincide throughout training).
   + GN converges faster with larger learning rates, but achieves best generalization when learning rate is small. Therefore good generalization with GN comes at the cost of slower convergence.
   + Finally, across all experiments, the WCD metric is a good proxy to the test loss.

Finally, the Appendix provides very detailed mathematical proofs and extra experimental results (e.g. use of the SiLU activation function instead of ReLU, experiments on MNIST).


**Strengths:**

- This is a beautifully written paper that is readable for non-super experts of this literature (as I am).
- The maths in Appendix are extremely rigorous and neat (I carefully checked the proofs).
- The conditions under which the derived theoretical guarantees hold are apparently less stringent than equivalent studies on GD (e.g. any activation function, initial weight not constrained to lie within a radius determined by the smallest singular value of the NTK).
- The experimental set up for the teacher-student study is sound and the RF baseline is extremely informative about what is going on.

**Weaknesses:**

I should state upfront that I am not an expert of this literature, so I may not be aware of the expected theoretical and experimental standards that lead to acceptance. This being said, I'm overall disappointed by the **experiments**:

- **Most importantly**: the MNIST experiments (last page of the appendix) seem to contradict the main findings that GN has to operate in small learning rate regime to work best and that GN would fail to optimize the last (linear) layer -- the gap between test loss and test loss after linear re-fitting is even greater for GD than GN! In spite of this glaring observation in stark contrast with the results appearing in the main, the authors simply do not comment upon this phenomenon at all (L.744-747).

- Apart from the choice of the model which sticks to the mean-field limit introduced in Section 3 (e.g. large number of hidden neurons, normalizing the model output by M), the theoretical and experimental parts are very disconnected. While I do understand that the intention was to cover optimization and generalization in a complimentary fashion (the former theoretically, the latter empirically), theory does not drive much the design of the experiments, and experiments are not crafted to validate the theory. I make some suggestions in the questions below.

- For the experiments: while I do understand that the choice of a one-hidden layer teacher-student setting seems to be a standard in this literature, it lies in between something that could be even more theoretically tractable with controllable analytical quantities (e.g. a toy quadratic optimization problem with known curvature) and something more realistic (e.g. multiple hidden layers model). So I'm twice frustrated that the theory hasn't been checked in an even simpler setting where everything is analytically well controlled, nor empirically tested against a model that is more complicated, to see if the proposed empirical prescriptions hold beyond this simple setting.

- I'm skeptic about the stopping criterion used for GN and worried that **it might skew all the experimental results**.

- In spite of all the rigor and clarity of the proposed study (which I do appreciate a lot), there is unfortunately no concrete practical prescription to use the GN algorithm. Even more so because the results on MNIST contradict those obtained in the teacher-student setting.

**Questions:**

Important questions/remarks:

- My most important remark (hilighted above) is that I suspect your MNIST experiments are not included in the main because they directly contradict the analysis performed on the regression experiments: GD and GN exhibit the exact same trends, GN works best with large learning rates and optimizes the last layer even better than GD. If you have time by the end of the rebuttal phase, I would be very happy if you could "fix" the experiments, or make sense of the observed discrepancies between the regression problem and MNIST.

- Could you please define precisely what you mean exactly by "blow-up"? You directly say in the main that if w is defined over a finite horizon ($T < \infty$), then it is said to "blow-up". Then you say that blow up happens when the minimal singular value of the NTK is zero. My impression is that there is not enough intuition in the main as to why this is the case. Looking at the appendix, my own intuition is that when $\sigma^\star(A_w)$ approaches zero, $J_w^\dagger$ diverges (Eq. 15 in the Appendix), and since the GN vector field is "roughly" $\Phi(w) \approx J_w^\dagger \cdot \nabla \mathcal{L}$, the GN vector field itself diverges. Is this intuition correct? Could you please include such intuition in the main to better define what "blow up" is precisely?

- You say in the main that "GN provably faster than GD". Could you please write down explicitly the global convergence rate of GD to be able to compare it directly with that of GN derived in Proposition 1 (even if it is a well-known result)? However, the comparison does not seem this straightforward when reading L.196-197. I'm just wondering how much writing GD global convergence rate would enlighten this part.

- I would have been very happy to see an empirical validation of Proposition 2 on a hand-crafted toy model, even simpler than the teacher-student setting, e.g. $\mathcal{L}=\frac{1}{2}(\theta - \theta_\star)^\top \cdot Q \cdot (\theta - \theta_\star)$? On such problems the properties of the spectrum of $\mathcal{L}$ could be well controlled. Taking $dim(\theta) = 2$ and visualizing trajectories in case of convergence, divergence, and showing the trajectory of GD as well would have been nice to see!

- L.223, you write: "The near-optimality condition on v0 can always be guaranteed by optimizing the second variable v while keeping u0 fixed". So why didn't you put it in practice in your own experiments? Especially because you observed that the last layer was not training so well. I'm not saying though that all GN experiments should have been performed with v-pretraining, but extra experiments could have been included. To make myself clearer, here is the gist of some experiment I would have been happy to see:
  - identify a failure mode of GN in the student-teacher setting -- i.e. the weight dynamics diverge.
  - show that by pre-training $v$, the problem vanishes (i.e. the weight dynamics converge to an optimum).
  - demonstrate it on both the teacher-student problem, and on more complex problem like MNIST.

- L. 297: why is the stopping criterion for GN lesser than that of GD ($K^{GD}=10^6$, $K^{GN}=1.5 \times 10^5$)? Is this because you observed blow-up? This choice is glaring in Fig. 2, bottom left panel and looks very strange: the GN learning trajectory stops at $10^5$ iterations and as the loss was gently decreasing, I would have been happy to see it go even lower! The bottom line is that the stopping criterion for GN looks quite arbitrary and cherry-picked. Why not using an $\epsilon-$relative-change-of-loss stopping criterion, namely: stop the simulation where the parameters have converged up to precision $\epsilon$?

- L.336: "this is unlike GD where larger step size yield better performing features". Looking closer at Fig. 1: indeed, the test loss decreases until $\lambda=10^{-3}$, but what happens beyond? Here again, the cut seems arbitrary. And as we expect, when looking at the blue curves (GD) Fig. 6 in the Appendix, the test loss for GD does increase beyond $\lambda=10^{-3}$. It is not a major issue, however I'm pointing out that the experimental analysis in the main is skewed again by arbitrary choices.

Minor questions/remarks:
- For non-experts like me: could you please an intuition why overparametrization (A) is defined this way? Perhaps this is something obvious for researchers of this community.
-  You say in the main that your theoretical results hold both in the kernel and feature learning regime. However, I do not understand why - here I am certainly missing some subtleties. Perhaps it would have been helpful to add background knowledge in the appendix about the exact definition of the feature and kernel limit. Is this about normalizing the model output by $1/\sqrt{M}$ (kernel) or $1/M$ (mean-field, as done in this paper)?
- In the same vein, I'm confused about whether the kernel/feature-learning *regime* is the same as (or related to) the kernel/mean-field *limit*. For the paper to be self-contained, it would be useful (at least for me) to add more background knowledge in the Appendix.
- L.207: could you explain the intuition?

**Limitations:**

The main limitation of the paper is the quality of the presented experiments, the fact that they are not consistent across two different learning problems, and that the results presented in the paper do not lead to a concrete prescription. For the sake of having the proposed work prescribe concrete takeaway prescriptions to use the GN algorithm in practice, I would recommend accept by the end of the rebuttal phase, once:
- the results you obtain on MNIST (on a single hidden layer architecture, consistently with the student-teacher setting) are consistent with the results presented in the main.
- the choice of the stopping criterion for GN is justified.

**POST-REBUTTAL UPDATE**: the authors have thoroughly addressed my concerns, so I increased my score to accept.

---

> ### Author Rebuttal · Authors · 2023-08-09
>
> Thank you for your extremely precise and outstanding review! We are very pleased to read that you found the paper well-written and that you took the time to carefully check the proofs.  We hope our answer clarifies all the remaining points.
>
> I. **Main points:**
>
>    1. **MNIST experiments:** We agree Figure 8 exhibits a different behavior than the experiment in the main. This is attributed to the training objective that is under-optimized for small step-size and thus ‘mechanically’ results in lower performance compared to larger step sizes. This would illustrate the trade-off between good generalisation vs computational cost due to smaller step sizes. Due to time constraints, we are unfortunately unable to provide further experiments on MNIST, but we agree that it would be valuable to strengthen those results.
>
>    2. **Connection between theory and experiments:**
>       - While Proposition 2 provides a global convergence result for the training loss, it says nothing about feature learning and is “happy” with the features staying near initialization as long as the loss reaches a global solution. The experiments illustrate a limitation of proposition 2 by showing that: It is not enough to globally optimize the training loss to get good generalization (as suggested by prop 2). Instead, the amount of feature learning during optimization is what essentially drives generalization performance.
>       - Pre-training the last layer to near optimality: Please refer to the section reserved to this point.
>
> 3. **Motivation for the 1 hidden-layer model:** This is one of the simplest intractable models for which the dynamics of GN can diverge and can have non-global solutions. We will clarify this in the text and in particular discuss simpler and more complex models as follows:
>    - Linear models with a strongly convex objective (like a quadratic problem) always converge to a global solution. The dynamics is indeed much simpler to analyze since the NTK matrix A_w remains constant by construction. In that case, there is no need for the initial condition of prop 2 and a similar convergence rate can be obtained. Moreover, on a quadratic problem GN converges to the exact solution in a single iteration. As this behavior is well documented, we decided not to include this case.
>    - More complex networks: We do not make claims on deeper/more complex networks as this would different analysis that goes beyond the scope of this paper. We believe this work is a first step towards studying GN for more complex models.
>
> 4. **Stopping criterion for GN:** Apologies for the confusion. We had a maximal time budget of 24h per job (we run 720 jobs in total) to control the overall resources allocated to this project. We are currently running GN with a longer budget. We expect it to decrease smoothly and will update the result as soon as we get it.
>    - In all cases, the time limit ensured the training loss (relative to its initial value) was below 10^{-5} and, in most cases, it was below 10-7  (see figure 2 right).
>    - Since the cost per iteration of GN is large, the time budget was exhausted at an error of 10^{-5} when using small step sizes for GN.
>
> 5. **Practical prescription:**
>    - A commonly used prescription for GN is to use the largest possible step size as it achieves faster convergence of the training objective. We show that this results in poor generalization (similar to RF).
>    - Instead, we show that smaller step sizes for GN allow better feature learning and are thus desirable but they come at a higher computational cost.
>    - Ultimately, there must be a tradeoff in the step size to achieve the best generalization at the lowest computational cost. An interesting research direction would be to quantify this optimal choice theoretically, but this is beyond the scope of this work.
>
> II. **Major questions:**
>
> 1. **Definition of “Blow-up”:** Apologies for the confusion, your intuition is correct and we will clarify in the text.
>
> 2. **Comparing convergence rates of GN and GD:** Thank you, we will provide the rates of GD for more clarity.
>    - For GD:   e^{-(mu sigma_min)/4 t} where sigma_min is the smallest eigenvalue of the NTK matrix A_w.
>    - For GN: for instance, when choosing H to be the identity (natural gradient) so that  L_H =\mu_H=1 and by choosing alpha = 1, the rate is: e^{- t mu }.
>
>    The eigenvalue sigma_min can be arbitrarily small, thus drastically slowing down the convergence rate of GD. This is not the case for GN.
>
> 3. **Pre-training the last layer to near optimality:**
> We will clarify that the RF baseline we provide is the extreme case where we train the last layer to exact optimality (instead of near optimality). In that case, the gradient vanishes and it is not possible to move away from the RF solution (which is a global one), so that feature learning is not possible. The intermediate case where the last layer is only near optimal can allow changing the inner weights to some extent but this gets worse as the last layer is closer to optimality.
>    - The failure mode of  GN (diverging dynamics): We have not observed the divergence of the dynamics empirically.
>
> 4. **Larger step-sizes** We noticed that larger values do not allow GD to converge to a minimizer (as you noted in Fig 6, you can see that the training loss remains large). Please note that this non-convergence for large step sizes is expected in optimization theory: there is a maximal step size beyond which GD with constant step size cannot converge, usually, this value is found by trial and error, which is what we did as best as our compute power allows.
>
> We will include clarifications for:
>    - Overparameterized model: we mean a model that can achieve zero training error on some fixed training set. The condition (A) ensures there exists a parameter value $(v_0,u_0)$ achieving 0 training error (which also explains the sentence in L207). We will include a simple proof to show that.
>    - NTK/mean-field regimes/limits.

---

> > ### Comment · Reviewer_c9sb · 2023-08-10
> > **Post-rebuttal update**
> >
> > Hi! Thank you for your kind answer. As per NeurIPS policy, I hereby acknowledge I thoroughly read your rebuttal.
> >
> >
> > **I. Main points**:
> >
> >    **1. MNIST experiments:** in spite of your answer, this seems to contradict your main claim in the paper that "GN achieves an improved generalization when using smaller learning rates". My straightforward suggestion would be to re-run these experiments **using more epochs** to balance the reduction of the learning rate and see if the expected behavior (that described in the main) is recovered.
> >
> >   **2. Connection between theory and experiments:** OK
> >
> >    **3. Motivation for the 1 hidden-layer model:** OK thank you for making this clearer, I realize my suggestion of the quadratic optimization toy problem was a naive suggestion.
> >
> >    **4. Stopping criterion for GN:** OK, thank you for this clarification. I'm sorry to read you had these limitations for the computation ressources, especially for the MNIST experiments, this is unfortunate. Looking forward to having your results, and hopefully by the end of the rebuttal phase!
> >
> >    **5. Practical prescription:** thank you for clarifying this. It would be great to make an addition along these lines, in the same very simple vein, along with your updated experimental results.
> >
> > **II. Major questions:**
> >
> >    **1. Definition of “Blow-up”:** OK.
> >
> >    **2. Comparing convergence rates of GN and GD:** OK.
> >
> >    **3. Pre-training the last layer to near optimality**: I do understand that the RF baseline brings the last layer to optimality. Is is also how you ran your MNIST experiments? If this is the case, my apologies for missing this. As to the divergence behavior: this would add an invaluable addition to the paper (if possible) to empirically investigate when divergence happens on tractable problems and see if it can be theoretically accounted for (e.g. analyzing the spectrum of the NTK as you mention it in the main).
> >
> >    **4. Larger step-sizes**: OK thanks for the clarification.
> >
> > My most important takeaway: **I am really ready to increase the score to firm accept if you re-run your experiments with a larger number of epochs and see if the expected behavior (that presented in the main) is recovered**. I do take into account the constraints you have on the compute cluster at your disposal, but could you really not run these experiments at all on a laptop running overnight? I'm thinking that running MNIST training experiments should be tractable with sufficient RAM on a laptop.

---

> > > ### Author Response · Authors · 2023-08-13
> > > **MNIST experiments**
> > >
> > > Dear reviewer,
> > >
> > > Thank you for taking the time to engage with the discussion and for encouraging us to improve the experiments.
> > > We have been re-running the MNIST experiments with a few modifications and we are pleased to announce that the new results are consistent with the rest of the paper. We will discuss the remaining points in a separate answer.
> > > Below, we provide a table of the results we obtained as well as a description of the experimental setup and comments on the results.
> > > Please let us know if you have any question about these results.
> > >
> > > 1. **Comments:**
> > >
> > > From the result table, we observe that:
> > >
> > >    - Test loss after linear-refit steadily increases with the lr in the case of GN while it tends to decrease in the case of GD.
> > >    - Hidden learning is strong in the case of GN: for large values of test and train losses, the test loss after linear refitting is as low as 52.4 for the smallest step-size. This is much less pronounced in case of GD.
> > >    - Note that for the largest step-sizes (1e2, 1e3) GD diverges.
> > >
> > > 2. **Setup:**
> > > Compared to the MNIST experiment of the submission:
> > >
> > >    - We train for 100000 iterations and set a stopping criterion when the training loss is below 1e-6.
> > >    - **We used a smaller initial std (0.001 instead of 1):** This allows us to be in a regime where GN and GD have different behaviour. Indeed, from Figure 1 (left) of the main, one can see that generalisation of GD and GN was similar for std =1, while there was a clear benefit for GN for smaller std. We recover this behaviour in the MNIST experiment.
> > >    - **Squared loss on the logits instead of the KL:** This choice allows us to get as close to the main experiment as possible. Besides, the  KL is not strongly convex while the L2 loss is, which is more consistent with the setup of the paper.
> > >
> > >
> > > 3. **Results:**
> > >
> > > | step-size                     | 1e-3   | 1e-2   | 1e-1 | 1    | 1e1  | 1e2   | 1e3   |
> > > |-------------------------------|--------|--------|------|------|------|-------|-------|
> > > | GN:  Train loss               | 330.   | 96.    | 15.5 | 1e-6 | 1e-6 | 1e-6  | 1e-6  |
> > > | GN:  Test loss (linear-refit) | 52.4   | 53.2   | 54.4 | 56.0 | 56.0 | 55.1  | 60.9  |
> > > | GN:  Test loss                | 337.14 | 134.11 | 66.9 | 56.5 | 56.4 | 55.6  | 60.9  |
> > > | GD:  Train loss               | 200.6  | 61.5   | 1e-6 | 1e-6 | 1e-6 | div | div |
> > > | GD:  Test loss (linear-refit) | 165.4  | 88.2   | 64.6 | 64.3 | 59.1 | div   | div   |
> > > | GD:  Test loss                | 208.9  | 88.3   | 64.8 | 64.4 | 59.2 | div   | div   |

---

> > > > ### Comment · Reviewer_c9sb · 2023-08-13
> > > > **MNIST experiments**
> > > >
> > > > Thank you, I have raised my score to accept!

---

> > > > > ### Author Response · Authors · 2023-08-14
> > > > > **Thank you**
> > > > >
> > > > > Dear reviewer,
> > > > >
> > > > > We wanted to express our sincere thanks for your review and for raising your score, which could make a big difference in the outcome.
> > > > > We will update the paper based on your suggestions which were instrumental in strengthening the paper.
> > > > >
> > > > > Thank you

---

### Official Review · Reviewer_6UDU · 2023-07-07

**Soundness:** 4 excellent
**Presentation:** 3 good
**Contribution:** 1 poor
**Rating:** 3
**Confidence:** 4

**Summary:**

The authors study the Gauss-Newton optimization algorithm in the overparameterized setting. They derive conditions for convergence of the Gauss-Newton algorithm with parameter-dependent damping in terms of the convexity of the loss, the smoothness of the loss Hessian, and the damping constant.

They then carry out an empirical study with a student-teacher, one hidden layer setup (overparameterized student), and show that Gauss Netwon in this setting requires small step size and small initialization to be successful. They also show that GN requires a larger number of steps but seems to induce more feature learning.

**Strengths:**

The idea of studying GN in the feature learning limit and trying to understand its implicit biases is an interesting one. The convergence analysis is new to my knowledge, and the experiments focus nicely on key issues of learning dynamics from second-order methods.

**Weaknesses:**

The algorithm as presented requires a parameter-dependent damping constant, which requires computation of the smallest non-zero NTK eigenvalues. These values are very difficult to compute in practice, and the algorithm is generally impractical. The theory consists of a convergence result for the aforementioned algorithm, and it’s not clear how relevant it is in practice.

Part of the issue with second order methods like GN are their impracticality in large model, large data settings. The experiments are not on any realistic datasets, and don’t suggest any promising ways forward for development of GN methods.

There may be connections to other mean field regimes which are not discussed in the text.

**Questions:**

What is the connection of the mean field analysis to the regimes from these two papers?

https://proceedings.neurips.cc/paper_files/paper/2022/hash/d027a5c93d484a4312cc486d399c62c1-Abstract-Conference.html
https://proceedings.mlr.press/v139/yang21c

**Limitations:**

Yes

---

> ### Author Rebuttal · Authors · 2023-08-09
>
> Thank you for your review. We are glad you find our study of the generalization and implicit bias of GN to be interesting. We believe the following clarifications should address all your concerns. If you still have concerns, we are more than happy if you could kindly share them as this would help us improve this work.
>
>
> 1. **The damping is optional:** Theoretical results in Proposition 2 cover the case without damping ($\alpha = 0$).  This is stated in L180 where we say "\alpha is non-negative". However, for more clarity, we will discuss the particular case when $\alpha =0$.  The proof does not rely on alpha>0. In our experiments, the smallest singular value comes for free after performing an SVD on a matrix of the size of a batch of data.
>
>  2. **Practicality and computational cost:** GN requires solving a system of  size N (N is the size of a mini-batch). More importantly, this does not require computing a matrix of size M (number of parameters). This is thanks to the Woodbury matrix identity as we discuss in L282-284. In our setting, we use a full batch, but using smaller batches is also possible and works well in practice see ref [11].
> In this work, we focus on the ideal algorithm GN without introducing additional biases or stochasticity in the estimation which can interfere with the interpretation of the results. This approach allows us to understand the ideal performance that a scalable approximation of GN can aim towards.
>
> 3. **Connection with mean field analysis:**
> Thank you for the reference, we will include it. As pointed out in their work, the mean-field regime they discuss is the same as the one studied in Chizat 2018, all of which we discussed in the introduction and L140-149. Note that, all these works consider the gradient flow dynamics while, here, we consider the dynamics of Gauss-Newton in the mean-field limit. Hence, the analysis is not directly applicable to GN, although the tools considered might be useful for future work.

---

> ### Author Response · Authors · 2023-08-13
> **We would be grateful if you engage with us during the discussion period**
>
> Dear reviewer,
>
> We thank you for your time.
> Since the authors/reviewers discussion period is still ongoing, we would be grateful if you give us the opportunity to engage in a constructive discussion about our submission.
>
> We believe we responded to the main three concerns you had about the paper:
>    - The damping is optional: the theory result holds even without damping.
>    - The computational cost can be easily reduced: thanks to Woodbury matrix identity and the use of mini-batch.
>    - The mean field limit is clarified: The paper you mentioned recovers the same mean-field limit we consider and which is equivalent to the one considered in [14,50].
>
> Based on these clarifications, do you still recommend rejection? Do you have any other doubts?
>
> We are grateful that two reviewers increased their scores based on the constructive discussions we had with them, including c9sb who now recommends an accept with a score of 7.
>
> We hope you can give us a chance to improve this work and clarify any concerns you might have.
>
> Thanks again for your time.

---

> > ### Comment · Reviewer_6UDU · 2023-08-15
> > **Response to author comments**
> >
> > I thank the authors for their comments.
> >
> > Regarding the damping: while I understand that proposition 2 does not require damping, it does require being near the optimal solution. I am more concerned with convergence rates as in proposition 1. I also don't quite understand how minibatch SVD gives good estimates of the smallest eigenvalues of the full NTK - large eigenvalues can sometimes be stochastically estimated in that way, small eigenvalues cannot.
> >
> > The issue I had with the computational cost was the fact that one needs to solve a dataset x dataset linear system. In the case presented here, with the toy dataset, the dataset size is 500 only so it is not too bad; on larger datasets it would be impractical. Thank you for pointing out that this method can in fact be minibatched for larger datasets. How does the effectiveness of the algorithm depend on batch size? For larger datasets, practical batch sizes can be small compared to the dataset size, and minibatches provide poor/biased estimators of the true Hessian/NTK. Does this affect the usefulness of GN? I still have concerns over this point, and feel that more extensive experiments are required.
> >
> > Thank you for your response about the reference as well.

---

> > > ### Author Response · Authors · 2023-08-16
> > > **Clarifications about the scope of the paper and scalabitlity**
> > >
> > > Thank you for your response and for further engaging with us in a discussion.
> > > Below, we clarify the remaining concerns starting by clarifying the scope of our work which is not about scalable approximations of GN, and then address each point in detail.
> > >
> > > 1. **Scope of the paper:**
> > >    - **This paper does not aim to address the scalability of GN**, many prior works provided scalable approximations with statistical guarantees for related methods (see for instance [4] which can fall into our proposed framework for GN).
> > >    - **This work rather asks the question:**  Does GN have good convergence and generalisation properties?
> > > The answer to this question is important because, if positive, it justifies research directions towards accurate and scalable approximations of GN.
> > >    - Our work suggests that GN indeed has good convergence and generalisation properties which is encouraging further research on making scalable approximations that retain these properties. It also highlights the Trade-off between optimization speed and generalisation that these approximations should maintain: Small step-sizes rather than larger ones favors feature learning rather than fitting the linear weights on random features.
> > >
> > >
> > > 2. **Effect of small mini-batch on the usefulness of GN:** This is undeniably an interesting question that is beyond the scope of this work (as discussed above).  There have been prior works studying the quality of the approximation on a mini-batch, for example for the Wasserstein natural gradient [4] (which is a particular case of the generalised GN we consider (different choice for the matrix H)). There, convergence rates for the approximation are provided and experiments on image datasets show positive results. However, since our work is not about scalable approximations to GN, we will make sure we do not make any claim about the matter in the paper.
> > >
> > > 3. **Estimating eigenvalues of the full NTK form mini-batch:**
> > >    - In our experiments, we computed the eigenvalue on the full batch of data. We never claimed that one can accurately estimate those from the SVD of a mini-batch NTK. We will make sure this is clear in the paper.
> > >    - That being said, there exist algorithms with statistical guarantees for estimating the smallest eigenvalues using mini-batches of data, and these date back to the seminal work of Oja and Karhunen, 1985 “On Stochastic Approximation of the Eigenvectors and Eigenvalues of the Expectation of a Random Matrix”.
> > >    - Finally, as discussed in the previous points, scalable approximations using mini-batch are beyond the scope of this work.
> > >
> > > 4. **“Regarding the damping: while I understand that proposition 2 does not require damping, it does require being near the optimal solution”:**
> > >    - We are unable to see the connection between the damping and initial condition being near the optimal solution, if you could please clarify what connection you have in mind, we can attempt to answer adequately.
> > >    - We also apologies if from the statement it can be inferred that the requirement on the initial condition can be perceived as a strong requirement. We have a discussion after proposition 2 in L221-224 explaining that such a requirement is in fact very easy to enforce but randomly initializing the hidden weights and optimizing the linear layer alone. We will make this clearer in the text.
> > >
> > > 5. **“Concern about the convergence rate of proposition 1”:** Could you please clarify what your concern is about this proposition, as it is not clear to us from your message? The rate obtained is a linear rate that has an improved conditioning compared to gradient descent and thus illustrates the advantage of using GN for faster optimization of the objective.
> > >
> > > We thank you again for your time and we will make sure to answer any further questions you might have.
> > >
> > >
> > > Best,
> > >
> > > The authors.

---

> > > > ### Author Response · Authors · 2023-08-18
> > > >
> > > > Dear reviewer,
> > > >
> > > > We thank you for your time and engagement with our work.
> > > > We wanted to know your thoughts about the answers we provided to your previous comment. In particular, if there are any remaining concerns that you might have.
> > > >
> > > > Thank you
> > > >
> > > > The authors

---

> > > > ### Comment · Reviewer_6UDU · 2023-08-18
> > > > **Response to clarifications**
> > > >
> > > > Thanks for the responses.
> > > >
> > > > In terms of the scope of the paper, I think convergence and generalization in _practical_ scenarios are the most important - there have been many second order methods which worked on small problems, but did not work well on larger ML/deep learning workloads. If the main contribution of the paper is the proofs, I still feel that the overall contribution is small.
> > > >
> > > > Regarding the small eigenvalue and convergence analysis: my question was mainly that the convergence rate does seem to depend on being able to regularize using the small eigenvalues. I re-iterate that in practice computing small eigenvalues accurately with small batches is not easy.
> > > >
> > > > In light of this, I will be keeping my review score.

---

> > > > > ### Author Response · Authors · 2023-08-19
> > > > >
> > > > > Dear reviewer,
> > > > >
> > > > > Thank you for your valuable feedback.
> > > > >
> > > > > We genuinely appreciate your time and effort in reviewing our work. It's evident from your comments that there might be some aspects of our chosen setting that require further clarification. We apologise for any confusion that may have arisen from our initial explanation.
> > > > >
> > > > > 1. **The setting of the paper:** is the full-batch GN for optimising over-parameterized one hidden layer in the mean-field regime.  In our introduction, we aimed to discuss the significance of the chosen setting as a foundation for our study. We will be happy to emphasise more the significance in the revised version based on the following points:
> > > > >
> > > > >    - **It is already a challenging problem:** The questions of convergence and generalisation of neural networks are challenging questions in general. Focusing on a full-batch setting is the first step in addressing this challenging problem. Many recent works restricted their attention to gradient descent in a full batch setting to study this problem. This work follows a similar objectif but for GN, since this method can potentially speeds up training, but its generalisation properties are not yet well understood.
> > > > >    - **Need to disentangle scalability issues from generalisation/convergence:** Without a good understanding of the ideal full-batch method it is hard to know what approximations are more likely to retain the essence of the method. Adding scalability considerations, are of course interesting ultimately, but introduces much more complications to an already hard problem.
> > > > >    - “There have been many second order methods which worked on small problems, but did not work well on larger ML/deep learning workloads.:” We believe this work studies GN in a setting that was not considered before (from the lens of the inductive bias of optimization method in the mean-field regime) and that is relevant to the theory of convergence and generalisation in deep learning. If the reviewer has in mind references that consider the same setting as we do, we will be grateful if they can point them to us.
> > > > >
> > > > > Therefore, we believe the considered setting to be relevant and important scientifically.
> > > > >
> > > > >
> > > > >
> > > > > 2. **” If the main contribution of the paper is the proofs, I still feel that the overall contribution is small.”** The main contribution is not limited to the proofs. The contributions as stated in the introduction they are twofold:
> > > > >    - The proof for global convergence of GN for two-layer networks in the mean-field regime.
> > > > >    - The empirical study of the inductive bias of GN which comes with a number of conclusions:
> > > > >       - Smaller steps-sizes yield better generalizaiton on full batch GN for over-parameterized networks in the mean-field regime
> > > > >       - Hidden learning of the features: the train and test loss can be large but GN with small step-size still learns good features as revealed by the linear-refitting.
> > > > >
> > > > >
> > > > > 3. **”my question was mainly the convergence rate does seem to depend on being able to regularize using the small eigenvalue”**: From your previous comment “Regarding the damping: while I understand that proposition 2 does not require damping”, we understood that our previous response already answered your question. We apologise if that was not the case. Please find further clarifications below:
> > > > >
> > > > >    - When the regularisation coefficient $\alpha$ is first introduced in L180, it is stated that it is a non-negative number. This means that $\alpha$ can be equal to $0$. In that case, no damping is used at all. We will make this more explicit.
> > > > >    - The statements of prop 1 and 2 are designed to work both with and without the regularisation. This is based on the initial definition of $\alpha$ which can take a value of $0$. It is not implied anywhere in the statement that $\alpha$ must be positive.
> > > > >    - The proofs in the appendix do not make any use of the regularisation anywhere. One can always take $\alpha=0$ in the proofs and they work in the same way.
> > > > >
> > > > > 4. **”I re-iterate that in practice computing small eigenvalues accurately with small batches is not easy."**
> > > > > 	Thank you for insisting on this point. We observed little difference when either using damping or not, i.e. setting alpha= 0 or 1 did not change the overall conclusions in terms of generalisation and convergence of GN.  That is because the smallest singular value was very small in practice. Since the damping is optional, the practicality of computing the smallest eigenvalue is irrelevant in our context. We will make this clear in the text.

---

### Author Response · Authors · 2023-08-21

Dear reviewers and AC,

We deeply appreciate your thorough discussions and invaluable insights that have significantly contributed to refining our work.

Your thoughtful feedback has enabled us to provide comprehensive clarifications that address your concerns. These clarifications will be seamlessly integrated into the main text, enhancing its quality.

We'd like to highlight two key points for your consideration:

1. **Reviewer 6UDU: Relevance of the setting: Why considering only full-batch GN, which is expensive computationally?**
We've shared a detailed response that underscores the scientific significance of our approach. Please find the response here: https://openreview.net/forum?id=8Oukmqfek2&noteId=FZ43YgUgwa

2. **Reviewers oHC4 and 2qbp: Technical and conceptual novelty in the convergence result of proposition 2.**
We've addressed the concerns of both reviewers separately, highlighting the unique technical and conceptual contributions:

   - For reviewer oHC4: https://openreview.net/forum?id=8Oukmqfek2&noteId=eJ3zQTKBad
   - For reviewer 2qbp: https://openreview.net/forum?id=8Oukmqfek2&noteId=8R1vJkitdM

We are confident that these clarifications enhance the understanding of our work and kindly encourage you to consider the potential for adjusted evaluations.

With gratitude,

The authors.

---

### Decision · Program_Chairs · 2023-09-21

**Decision:**

Accept (poster)

**Comment:**

This work studies the global convergence and generalization properties of the Gauss-Newton (GN) method when optimizing one-hidden layer networks in the over- parameterized regime. This is a welcome contribution, as the literature focused more on gradient descent. Following the discussion period, the authors went through lots of efforts to clarify the paper.